# Integration of clinical, pathological, radiological, and transcriptomic data improves prediction for first-line immunotherapy outcome in metastatic non-small cell lung cancer

Nicolas Captier [1,2] ✉, Marvin Lerousseau [2,3], Fanny Orlhac [1], Narinée Hovhannisyan-Baghdasarian [1], Marie Luporsi [1,4], Erwin Woff [1,5], Sarah Lagha [6], Paulette Salamoun Feghali[6], Christine Lonjou [2], Clément Beaulaton [7], Andrei Zinovyev [8], Hélène Salmon [9], Thomas Walter [2,3,10], Irène Buvat [1,10], Nicolas Girard [6,10] & Emmanuel Barillot [2,10] ✉

Immunotherapy is improving the survival of patients with metastatic non-small cell lung cancer (NSCLC), yet reliable biomarkers are needed to identify responders prospectively and optimize patient care. In this study, we explore the benefits of multimodal approaches to predict immunotherapy outcome using multiple machine learning algorithms and integration strategies. We analyze baseline multimodal data from a cohort of 317 metastatic NSCLC patients treated with first-line immunotherapy, including positron emission tomography images, digitized pathological slides, bulk transcriptomic profiles, and clinical information. Testing multiple integration strategies, most of them yield multimodal models surpassing both the best unimodal models and established univariate biomarkers, such as PD-L1 expression. Additionally, several multimodal combinations demonstrate improved patient risk stratification compared to models built with routine clinical features only. Our study thus provides evidence of the superiority of multimodal over unimodal approaches, advocating for the collection of large multimodal NSCLC datasets to develop and validate robust and powerful immunotherapy biomarkers.

Anti PD-1/PD-L1 immunotherapy with or without chemotherapy is the current standard first-line therapy for metastatic non-small cell lung cancer (NSCLC) without actionable oncogene alterations and without contraindications to PD-1/PD-L1 inhibitors[1]. Several clinical trials have indeed demonstrated significantly improved Overall Survival (OS) and Progression-Free Survival (PFS) with immunotherapy in comparison to chemotherapy alone[2–6]. Nevertheless, half of the patients do not present a radiological response to immunotherapy, and the duration of response remains highly variable from one patient to another (ranging from 1.1 to 18 months for patients treated with first-line immunotherapy + chemotherapy)[3]. Ultimately, the number of patients with long-term survival is limited. There is thus a critical need for

biomarkers that can predict treatment response accurately. These biomarkers will pave the way to better personalize the treatment strategy – immunotherapy as single agent for patients with predicted prolonged survival, combination with chemotherapy or other agents for patients with predicted poor response and survival -, to customize the follow-up and assess adequately treatment sequences.

Machine learning approaches have recently shown their potential to leverage data collected before treatment initiation, including clinical[7,8], radiological[9,10], anatomopathological[11,12], or transcriptomic information[13,14], and develop robust prognostic and predictive models that could outperform approved univariate biomarkers such as PD-L1 expression[15]. Promising results have fostered the exploration of multimodal approaches to combine all the diverse aspects of the disease that these different modalities probe. Yet, evidence of the superiority of multimodal over unimodal biomarkers[16] remains limited, possibly due to challenges in gathering comprehensive multimodal cohorts. Therefore, there is a pressing need for new studies involving large and homogeneous NSCLC multimodal cohorts to fully explore the benefits of multimodality and design strategies to address the challenges associated with integrating multimodal data.

In this study, we conduct a thorough comparison of unimodal and multimodal approaches for predicting the outcome of metastatic NSCLC patients undergoing first-line immunotherapy. Using a new multimodal cohort of 317 patients—including clinical data, PET/CT scans, digitized pathological slides, and bulk RNA-seq data—we demonstrate the superiority of multimodal approaches across the majority of the explored predictive algorithms and integration strategies. Mapping each modality to a set of interpretable features, we also identify the most influential factors for immunotherapy outcomes and explore their complementarity. These results could guide future research, fostering efforts to collect and analyze large multimodal cohorts, ultimately leading to the development and validation of a new generation of multimodal biomarkers that could transform NSCLC patient care.

## Results

### Clinical characteristics of patients with metastatic NSCLC

We identified 317 NSCLC patients treated at Institut Curie, who met the inclusion criteria: patients with histologically proven advanced NSCLC who received anti-PD-(L)1 immunotherapy, specifically pembrolizumab, as their first-line treatment. Immunotherapy was administered either as a standalone treatment for patients with a PD-L1 expression greater than 50% or in combination with chemotherapy, regardless of the PD-L1 expression, as per clinical practice guidelines[1]. PD-L1 expression was evaluated by immunohistochemistry (Sp263 and QR1 assays), with the Tumor Proportion Score (TPS) representing the percentage of tumor cells exhibiting membrane PD-L1 staining. The patients who received pembrolizumab as monotherapy were treated between October 2017 and January 2023 while those who received pembrolizumab combined with chemotherapy were treated between July 2019 and January 2023. The clinical characteristics of the multimodal cohort are detailed in Table 1.

Median OS and PFS were respectively 723 days (95% CI [446–987]) and 301 days (95% CI [145–598]) for the patients treated with immunotherapy alone, and 763 days (95% CI [576-NR]) and 290 days (95% CI [241–372]) for the patients treated with a combination of immunotherapy and chemotherapy (Fig. 1A). Interestingly, no significant difference was observed between the two treatment groups for OS (log-rank $p$-value = 0.44, Fig. 1B), even for the patients with PD-L1 expression greater than 50% only (Supplementary Fig. s1). We observed that for PFS, the immunotherapy + chemotherapy group had fewer early progressors, although this was compensated by an increase in late progressors compared to the immunotherapy-only group (Fig. 1A).

### Standard univariate biomarkers show limited predictive power

PD-L1 expression was able to stratify patients, with significant differences in PFS and OS in patients with negative expression (< 1%) from those with positive expression (≥ 1%) (Fig. 1B and Supplementary Fig. s2). However, it yielded a mild performance as a univariate biomarker for patient survival (C-index OS = 0.54, bootstrap 95% CI [0.51–0.57], permutation $p$-value = 0.014, $n$ = 298). Besides, no significant performance was observed when using PD-L1 expression as a continuous score, where negative expressions were replaced by 0%, and the score was calculated as 100 - TPS (C-index OS = 0.53, bootstrap 95% CI [0.48–0.58], permutation $p$-value = 0.104, $n$ = 295). Other standard clinical biomarkers, such as the Tumor Mutational Burden (TMB) or Tumor Infiltrating Lymphocytes (TILs) − with TILs being semi-quantitatively assessed on routine pathological sections without any cutoff − did not exhibit significant association with patient outcome (Fig. 1C and Supplementary Fig. s3).

### Collection of multiple baseline modalities to predict immunotherapy outcome

Clinical information from routine care, [18F]FDG-PET/CT scans, digitized pathological slides from the initial diagnosis, and bulk RNA-seq profiles from solid biopsies were collected at baseline. For each data modality, we first selected and computed several hand-crafted features to serve as input for both unimodal and multimodal predictive models, including 30 clinical features, 30 radiomic features, 134 pathomic features, and 34 transcriptomic features. We then leveraged this multimodal dataset to conduct an extensive comparison of the performance of unimodal and multimodal approaches using a 10-fold cross-validation scheme applied to the entire cohort and repeated 100 times (Supplementary Fig. s4).

237 out of the 317 patients had at least one missing modality (Fig. 1D). To ensure a fair comparison of all possible modality combinations, we, therefore, restricted the evaluation of prediction performance to the 80 patients with a complete multimodal profile (i.e., collecting the predictions of these 80 patients only, from the test sets of the cross-validation scheme applied to the whole cohort; Supplementary Fig. s4). Log-rank tests indicated no significant differences between the survival distributions of patients with missing modalities and those with available modalities (Supplementary Fig. s5).

### Comparison of unimodal performances across multiple prediction tasks

We first evaluated the predictive value of each modality individually. This evaluation involved predicting risk scores for time-to-event outcomes (OS and PFS) and classifying patients into two groups: those who would die within one year of treatment and those who would not (1-year death), or those who would experience disease progression before 6 months of treatment and those who would not (6-month progression). We focused on two standard Machine Learning approaches that are well-suited for datasets with modest numbers of samples[17]: linear methods (logistic regression and Cox regression with elastic net penalties) and tree ensemble methods (Random Survival Forest[18] and gradient-boosted tree[19] algorithms). The four modalities exhibited varying degrees of predictive power for patient outcome, with the RNA modality standing out for the prediction of 1-year death (AUC = 0.75 ± 0.04 (± 1std); Table 2 and Supplementary Table s1). PFS and 6-month progression predictions were more challenging than OS and 1-year death predictions. Except for pathological data, all modalities yielded greater performance (using either linear or tree ensemble algorithms) in predicting OS and 1-year death compared to PFS and 6-month progression. Across all modalities and models, the highest scores achieved were a C-index of 0.59 (± 0.02) for RNA modality in predicting PFS and an AUC of 0.61 (± 0.03) for clinical, pathomic, or RNA modality in predicting 6-month progression.

**Table 1 | Clinical characteristics of the multimodal cohort and the subset of patients with a complete multimodal profile**

| Clinical characteristics | | Multimodal cohort (n = 317) | Immuno + chemo (n = 196) | Immuno alone (n = 121) | Subset with all modalities (n = 80) | Statistical comparison (80 vs 237) |
|---|---|---|---|---|---|---|
| Age – median (range) | | 66 (33-92) | 64 (33-84) | 69 (40-92) | 64 (37-82) | $p_{val}^1$ = 4.9e-3 |
| Sex - n (%) | Men | 189 (60) | 113 (58) | 76 (63) | 46 (57) | $p_{val}^2$ = 6.0e-1 |
| | Women | 128 (40) | 83 (42) | 45 (37) | 34 (43) | |
| 1st line therapy – n (%) | Pembrolizumab + chemotherapy | 196 (62) | 196 (100) | – | 55 (69) | $p_{val}^2$ = 9.2e-2 |
| | Pembrolizumab | 121 (38) | – | 121 (100) | 25 (31) | |
| Histology - n (%) | Adenocarcinomas | 232 (73) | 152 (77) | 80 (66) | 54 (68) | $p_{val}^2$ = 2.8e-2 |
| | Squamous cell carcinomas | 44 (14) | 17 (9) | 27 (22) | 13 (16) | |
| | Other subtypes/not available | 41 (13) | 27 (14) | 14 (12) | 13 (16) | |
| PD-L1 expression – n (%) | ≥ 50% | 163 (51) | 49 (25) | 114 (94) | 42 (52) | $p_{val}^2$ = 6.2e-1 |
| | 1–49% | 82 (26) | 78 (40) | 4 (3) | 23 (29) | |
| | Negative | 56 (18) | 56 (29) | 0 (0) | 11 (14) | |
| | Not available | 16 (5) | 13 (6) | 3 (3) | 4 (5) | |
| Smoking status – n (%) | Current/former | 287 (91) | 180 (92) | 107 (88) | 71 (89) | $p_{val}^2$ = 6.9e-1 |
| | Never | 29 (9) | 15 (8) | 14 (12) | 8 (10) | |
| | Not available | 1 (<1) | 1 (<1) | 0 (0) | 1 (1) | |
| Performance status – n (%) | ECOG 0/1 | 244 (77) | 158 (81) | 86 (71) | 71 (89) | $p_{val}^2$ = 1.5e-3 |
| | ECOG ≥ 2 | 36 (11) | 14 (7) | 22 (18) | 1 (1) | |
| | Not available | 37 (12) | 24 (12) | 13 (11) | 8 (10) | |
| TILs – n (%) | Positive | 159 (50) | 82 (42) | 77 (64) | 49 (61) | $p_{val}^2$ = 2.8e-2 |
| | Negative | 18 (6) | 9 (5) | 9 (7) | 3 (4) | |
| | Not available | 140 (44) | 105 (53) | 35 (29) | 28 (35) | |
| Median Overall Survival – days (95% CI) | | 756 (592–910) | 723 (446–987) | 763 (576-NR) | 846 (650-NR) | $p_{val}^3$ = 2.7e-1 |
| Median Progression Free Survival – days (95% CI) | | 296 (241–372) | 301 (145–598) | 290 (241–372) | 386 (275–711) | $p_{val}^3$ = 1.6e-2 |

*ECOG* Eastern Cooperative Oncology Group, *TILs* Tumor-Infiltrating Lymphocytes, *NR* Not Reached, $p_{val}^1$ Welch's *t* test *p*-value - $p_{val}^2$ Chi-squared test p-value - $p_{val}^3$ Log-rank test *p*-value.
The results of the statistical comparison between the subset of patients with a complete profile and the rest of the cohort, using two-sided Welch's *t* tests, one-way Chi-squared tests, and Log-rank tests, are presented in the last column.

## Feature importance analyses highlight relevant clinical and transcriptomic features

We first investigated feature importance for the prediction of OS and 1-year death, providing insights into the information learned by each unimodal model (see Methods). Notably, it revealed that clinical models consistently learned that patients with a low level of serum albumin, a negative PD-L1 status (i.e., TPS < 1%), or abundant circulating neutrophils were more likely to have a poor prognosis (Fig. 2A). This analysis also highlighted the transcriptomic features that the RNA models used to predict OS and 1-year death (Fig. 2B). RNA models consistently associated an abundance of dendritic cells (DC), as scored by the MCP-counter method[20], or a high expression of NTRK1 gene with a good prognosis while they associated high expression of NRAS and KRAS genes with a poor prognosis. Interestingly, among the 13 consensus transcriptomic features identified with feature importance analysis (see Methods), only 3 exhibited significantly different values between biopsy sites in the 84 patients for whom this information was available (Supplementary Fig. s6), suggesting that the other 10 may be used independently of the biopsy location. Radiomic models primarily focused on information related to the Total Metabolic Tumor Volume (TMTV) as well as to the total metabolic volume of extra-thoracic metastases (Supplementary Fig. s7). Lastly, the interpretation of pathomic models unveiled features that encoded the proportion of inflammatory cells within the biopsy sections as well as their spatial organization (Supplementary Fig. s8).

We then turned to feature importance analysis of unimodal models for the prediction of PFS and 6-month progression. Notably, it confirmed that a high expression of the NTRK1 gene and an abundance of dendritic cells was associated with a favorable prognosis since they were also ranked among the top ten most important transcriptomic

features for multivariate predictions and showed significant univariate association with both PFS and 6-months progression (Supplementary Fig. s9A). Similarly, it highlighted the favorable influence of positive PD-L1 expression or a high level of serum albumin on the prognosis of multivariate models and the negative influence of a high neutrophils-to-lymphocytes ratio (Supplementary Fig. s9B). Finally, this analysis showed that, similarly to OS and 1-year death prediction, radiomic models were predominantly driven by the TMTV (Supplementary Fig. s10A).

The consensus important features (see Methods) identified for OS and 1-year death predictions, along with those for PFS and 6-month progression predictions exhibited mild to low inter-modal correlations (Supplementary Fig. s11), suggesting that the different collected modalities may capture distinct aspects of each patient's condition and response to therapy.

## Late fusion of unimodal predictors improves the prediction of immunotherapy outcome

We then developed multimodal predictors with the hypothesis that multimodal data would provide richer and more comprehensive information. We first applied late fusion as a baseline strategy for integrating all the modalities into multimodal predictors of OS, 1-year death, PFS, and 6-month progression (Supplementary Fig. s4). Late fusion consists of averaging the predictions of each individual unimodal predictor. We tested every possible combination of two to four modalities for each predictive task using both linear and tree ensemble algorithms. The late fusion of tree ensemble models improved the prediction of patient outcomes across both classification and survival tasks (Fig. 3 and Supplementary Fig. s12). Specifically, for 1-year death, the combination of predictions from clinical, RNA, and radiomic

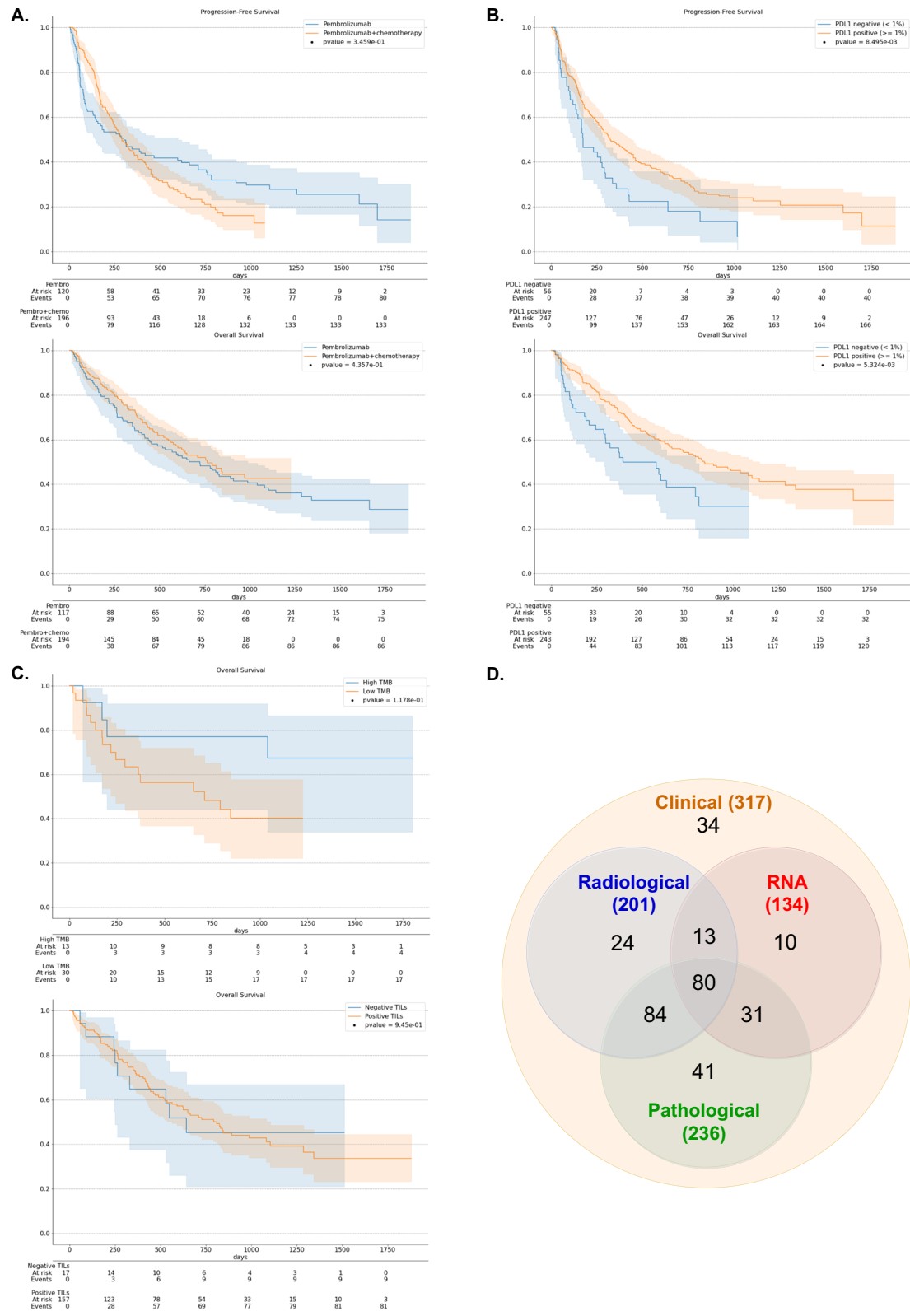

models demonstrated the highest performance (AUC = 0.81 ± 0.03) while, for OS, the combination of predictions from clinical and RNA models performed best (C-index = 0.75 ± 0.01). For both OS and 1-year death, paired-permutation tests confirmed the significantly higher AUC and C-index for the combined model compared to clinical-only, radiomics-only, and pathomics-only models (Supplementary Fig. s13). For PFS, the combination of predictions from clinical, RNA, pathomic, and radiomic models yielded the best performance while, for 6-month

progression, it was the combination of predictions of clinical, RNA, and pathomic models. However, the performance of these two combinations was not significantly different from those of unimodal models with paired-permutation tests. The late fusion of linear models performed better than tree ensemble models for the prediction of 6-month progression only, with a combination of clinical, pathomic, and RNA predictions yielding an AUC of 0.67 (± 0.03) (Supplementary Fig. s14). This can be explained by the greater

**Fig. 1 | Survival of NSCLC patients and Venn diagram summarizing the multimodal cohort. A** OS and PFS Kaplan-Meier survival curve (solid lines) for the whole NSCLC cohort (*n* = 311 for OS and *n* = 316 for PFS) with a 95% confidence interval (shaded areas). Patients are stratified with respect to their first-line therapy, either pembrolizumab alone or pembrolizumab + chemotherapy. Log-rank *p*-values are reported to characterize the separation of the survival curves. **B** OS and PFS Kaplan-Meier survival curves (solid lines) with 95% confidence interval (shaded areas) and log-rank *p*-values for the patients with available PD-L1 expression (*n* = 295 for OS and *n* = 300 for PFS). Patients are stratified with respect to their PD-L1 status (positive vs negative). **C** OS Kaplan-Meier survival curves (solid lines) with 95% confidence interval (shaded areas) and log-rank *p*-values for the 43 patients with available TMB and the 174 patients with available TILs status. For the TMB, patients are stratified with a threshold of 15 mutations per megabase (see Methods). For TILs, patients are stratified with respect to their positive vs negative TILs status. **D** Overview of the multimodal cohort with a Venn diagram. The four data modalities and their intersections are represented (i.e., PET/CT images, clinical data, pathological slides, and bulk RNA-seq profiles). Source data are provided as a Source Data file.

**Table 2 | Unimodal performance for the prediction of OS, 1-year death, PFS, and 6-month progression**

| Target (number of patients) | | OS (*n* = 79) | 1-year death (*n* = 77) | PFS (*n* = 80) | 6-month progression (*n* = 75) |
|---|---|---|---|---|---|
| Metric | | C-index | AUC | C-index | AUC |
| **Clinical** | Tree ensembles | 0.67 ± 0.01* | 0.59 ± 0.05 | 0.56 ± 0.02 | 0.58 ± 0.04 |
| | Linear | 0.60 ± 0.02* | 0.73 ± 0.02* | 0.53 ± 0.03 | **0.61 ± 0.03*** |
| **Radiomics** | Tree ensembles | 0.61 ± 0.02* | 0.62 ± 0.04 | 0.57 ± 0.01 | 0.56 ± 0.05 |
| | Linear | 0.61 ± 0.02* | 0.47 ± 0.03 | 0.55 ± 0.02 | 0.48 ± 0.04 |
| **Pathomics** | Tree ensembles | 0.59 ± 0.02 | 0.54 ± 0.05 | 0.56 ± 0.02 | 0.58 ± 0.06* |
| | Linear | 0.58 ± 0.02 | 0.56 ± 0.03 | 0.51 ± 0.02 | **0.61 ± 0.03*** |
| **RNA** | Tree ensembles | **0.69 ± 0.02*** | **0.75 ± 0.04*** | 0.57 ± 0.02 | 0.60 ± 0.04* |
| | Linear | 0.58 ± 0.02 | 0.65 ± 0.03 | **0.59 ± 0.02*** | **0.61 ± 0.03** |

*one-sided permutation *p*-value ≤ 0.05 (exact *p*-values are reported in Supplementary Table s1).

Unimodal performance of each data modality for the prediction of OS, 1-year death, PFS, and 6-month progression with linear and tree ensemble algorithms (mean ± std over the 100 cross-validation schemes). The best performances for each column are highlighted in bold. Source data are provided as a Source Data file.

performance of unimodal models with linear approaches for 6-month progression prediction, underscoring that the performance of late fusion combinations strongly depends on the performance of their unimodal components.

To further compare late fusion multimodal models with unimodal ones, we computed the marginal contribution of each modality to the final multimodal prediction for each patient. We focused on the best-performing model that combined clinical, radiomic, and RNA tree ensemble models for 1-year death prediction (Fig. 3). For several patients, the different modalities did not influence the multimodal prediction in the same direction (Fig. 4A). Notably, in 26% of the cases (20/77), the RNA modality's contribution was discordant with the final multimodal prediction. Among these discordant cases, one-third (6/20) were correctly influenced by the RNA modality but misclassified by the multimodal model, while two-thirds (14/20) were negatively influenced by the RNA modality but correctly classified by the multimodal model, with the radiomic and clinical modalities guiding the prediction towards the correct outcome (clusters 1&2). Analyzing the feature importance for the 14 patients where the multimodal prediction was correct despite the negative RNA contribution revealed that features from different modalities provided opposing information, balancing each other to guide the multimodal prediction in the correct direction (Fig. 4B). For instance, in some cases, high expression of NRAS gene negatively influenced the prediction, but incorporating clinical and radiomic information—such as elevated serum albumin level or high spleen metabolism—helped achieve a correct prediction. Overall, the three fused modalities exhibited diverse behaviors, with weak correlations between their unimodal predictions (Fig. 4C). Averaging their decisions impacted the predicted outcomes for several patients—not just isolated cases—and improved overall performance.

**Benchmark of integration strategies reveals a consistent benefit of multimodal approaches**

We compared the late fusion approach with early fusion (Supplementary Fig. s4). The baseline early fusion approach consists of concatenating the features from the different modalities and using these concatenated vectors as input to a single predictor. For binary classification tasks, we also re-implemented and tested an attention-based fusion approach known as DyAM[16], which was recently applied to NSCLC multimodal data. Early fusion and DyAM models were trained both without and with prior univariate feature selection to balance the dimensions of the different modalities (see Methods). The comparison of these different integration strategies for predicting OS, 1-year death, PFS, and 6-month progression did not identify a single best strategy (Fig. 5). The late fusion of tree ensemble models yielded the best performance for the prediction of OS and 1-year death, while the early fusion of tree ensemble models and the DyAM model, both with prior univariate feature selection, outperformed the other strategies for PFS and 6-month progression prediction, respectively. This comparison demonstrated the potential of multimodal approaches to enhance unimodal predictions, as for each prediction task the majority of integration strategies resulted in multimodal combinations that outperformed the best unimodal models. Furthermore, this comparison highlighted modalities that were consistently involved in the best multimodal combinations across the different integration strategies, particularly for 1-year death and 6-month progression prediction. For 1-year death, the integration strategies that outperformed the best unimodal model with their optimal combination combined clinical (5/7 strategies), RNA (7/7 strategies), and radiomic (3/7 strategies) modalities. For 6-month progression, they combined clinical (7/8 strategies), pathomic (7/8 strategies), and RNA (8/8 strategies) modalities. The combination of clinical and RNA modalities also performed best for OS prediction, while for PFS prediction, it was the combination of clinical, pathomic, and RNA. Lastly, the superiority of multimodal approaches was confirmed when comparing the average performance of the different integration strategies across all possible combinations of one, two, three, and four modalities (Fig. 6 and Supplementary Fig. s15). Indeed, the average performance at a fixed number of modalities increased with the number of integrated modalities for every strategy and every prediction task (except for the early fusion with a linear model and no prior feature selection). Paired sample *t* tests showed that multimodal combinations (involving two, three, or four

 

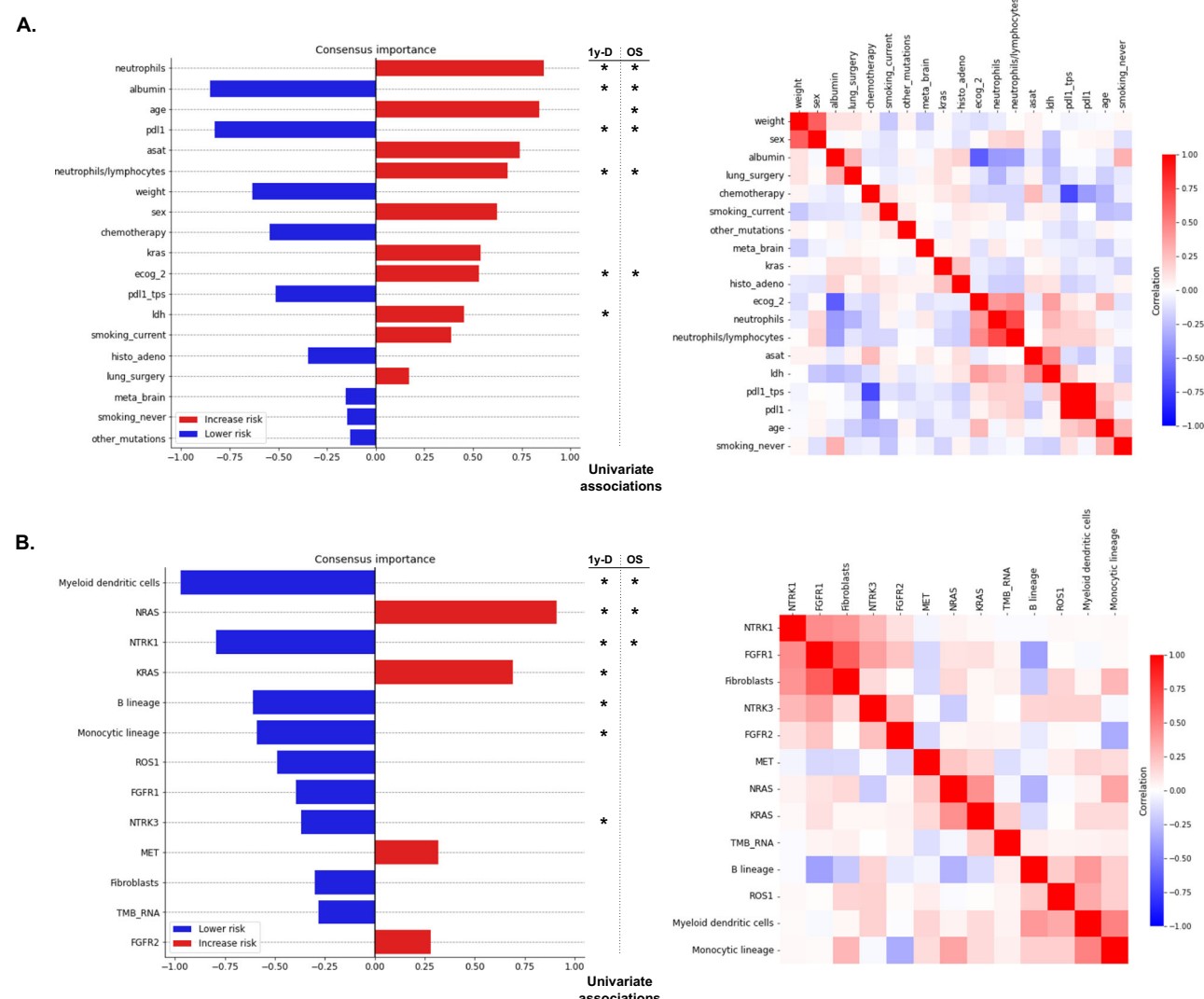

**Fig. 2 | Feature importance ranking for the prediction of overall survival, for clinical and transcriptomic modalities.** Feature importance ranking was obtained by aggregating the SHAP values collected from both tasks (OS and 1-year death) and both approaches (linear and tree ensemble methods) (see Methods). Features that were significantly associated with 1-year death (one-sided permutation test with univariate AUCs) after Benjamini-Hochberg (BH) correction ($\alpha = 0.05$) are shown with a * on the left side, while features that were significantly associated with OS (one-sided permutation test with univariate C-index) after BH correction are annotated with a * on the right side. * corresponds to an adjusted *p*-value below 0.05. **A** Consensus feature importance ranking for the clinical data modality (left) and heatmap of correlations between consensus clinical features (right). Correlations were evaluated by Spearman correlation coefficients (for continuous feature vs continuous feature), AUCs rescaled to [−1, 1] (for continuous feature vs binary categorical feature), or Matthews correlation coefficient (for binary categorical feature vs binary categorical feature). **B** Consensus feature importance ranking for the RNA data modality (left) and heatmap of Spearman correlations between consensus RNA features (right). Source data are provided as a Source Data file.

modalities) consistently led to performance improvements compared to unimodal models. The performances of all the multimodal models are detailed in the supplementary materials (Supplementary Figs. s14, s16–23).

We also explored whether the observed multimodal benefit depended on our initial selection of features within each modality. We focused on the transcriptomic modality, which demonstrated the highest unimodal performance (Table 2), and assessed whether our multimodal models could outperform any transcriptomic signature, not just the one derived from the initially selected features. For each predictive task, we applied the same cross-validation schemes as before and compared the predictive performance of the best multimodal model with 36 transcriptomic signatures previously associated with immunotherapy in the literature (see Methods, Supplementary Table s2). The best multimodal model outperformed all transcriptomic signatures, except for the prediction of 6-month

progression, where it outperformed 33 out of 36 signatures (Fig. 7). Interestingly, for OS and 1-year death, our best unimodal model ranked among the top two transcriptomic signatures, whereas for PFS and 6-month progression, it did not rank within the top ten.

### Multimodal predictions demonstrate improved patient stratification for OS

Kaplan-Meier analysis showed that the predictions of multimodal models, integrating clinical data with other modalities when available, effectively stratified patients' OS. After adjusting the log-rank *p*-values, 93% of all the combinations across the different prediction tasks (i.e., 328/352) exhibited significant differences between the survival distributions of their low-risk and high-risk groups (Fig. 8A and Supplementary Fig. s24). Notably, 74% of the combinations (i.e., 260/352) yielded a lower *p*-value than the binary PD-L1 status (log-rank *p*-value = 0.0025, *n* = 265). For each model, low-risk and high-risk

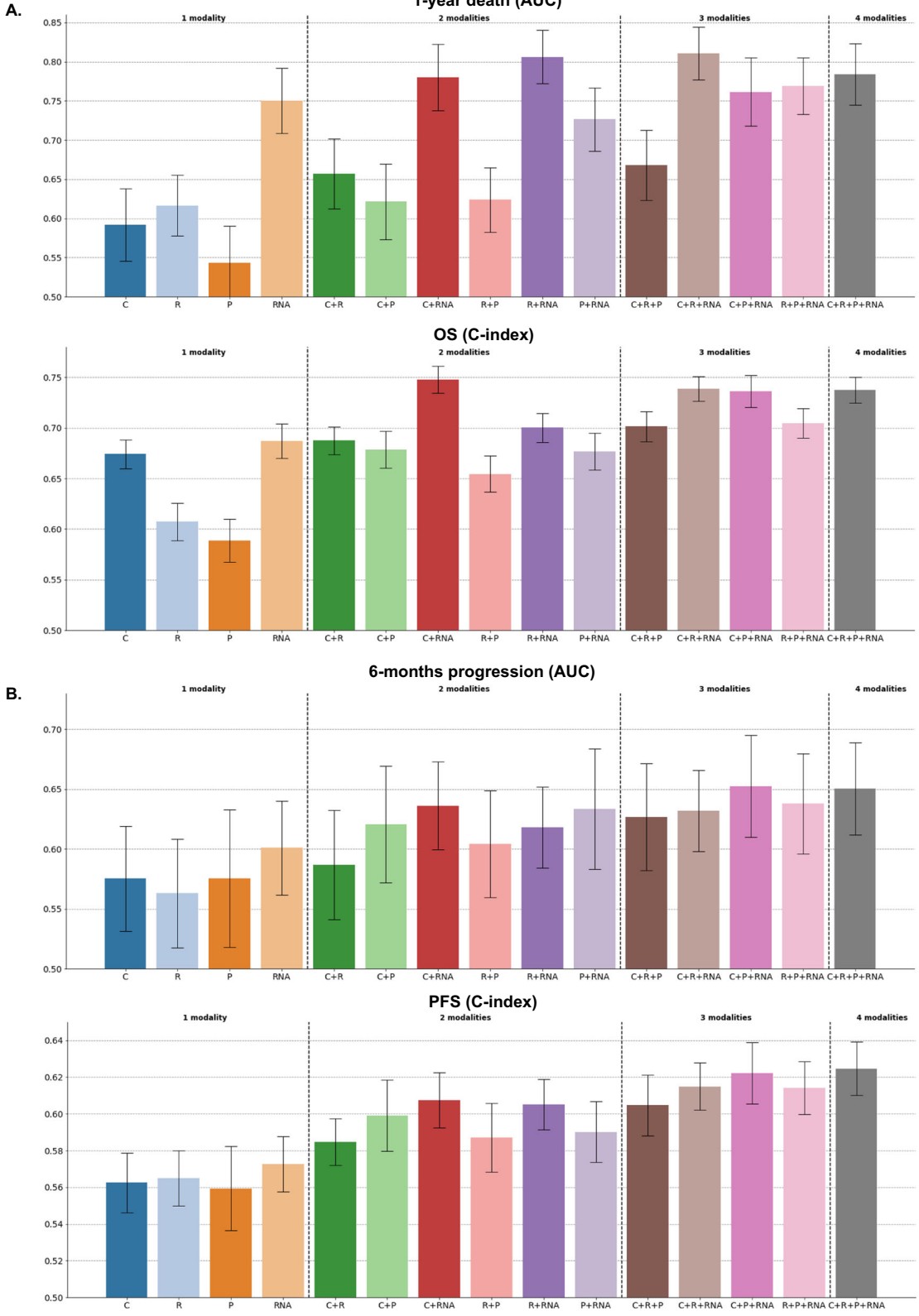

**Fig. 3 | Performance of all the possible multimodal combinations, with a late fusion strategy and tree ensemble methods.** The bar height corresponds to the performance metric (either ROC AUC or C-index) averaged across the 100 cross-validation schemes, and the error bar corresponds to ±1 standard deviation, estimated across the 100 cross-validation schemes. **A** ROC AUCs associated with the prediction of 1-year death with XGBoost algorithms (top) and estimated with n = 77 patients. C-indexes associated with the prediction of OS with Random Survival Forest algorithms (bottom) and estimated with n = 79 patients. **B** ROC AUCs associated with the prediction of 6-month progression with XGBoost algorithms (top) and estimated with n = 75 patients. C-indexes associated with the prediction of PFS with Random Survival Forest algorithms (bottom) and estimated with n = 80 patients. * C: clinical, R: radiomic, P: pathomic, RNA: transcriptomic. Source data are provided as a Source Data file.

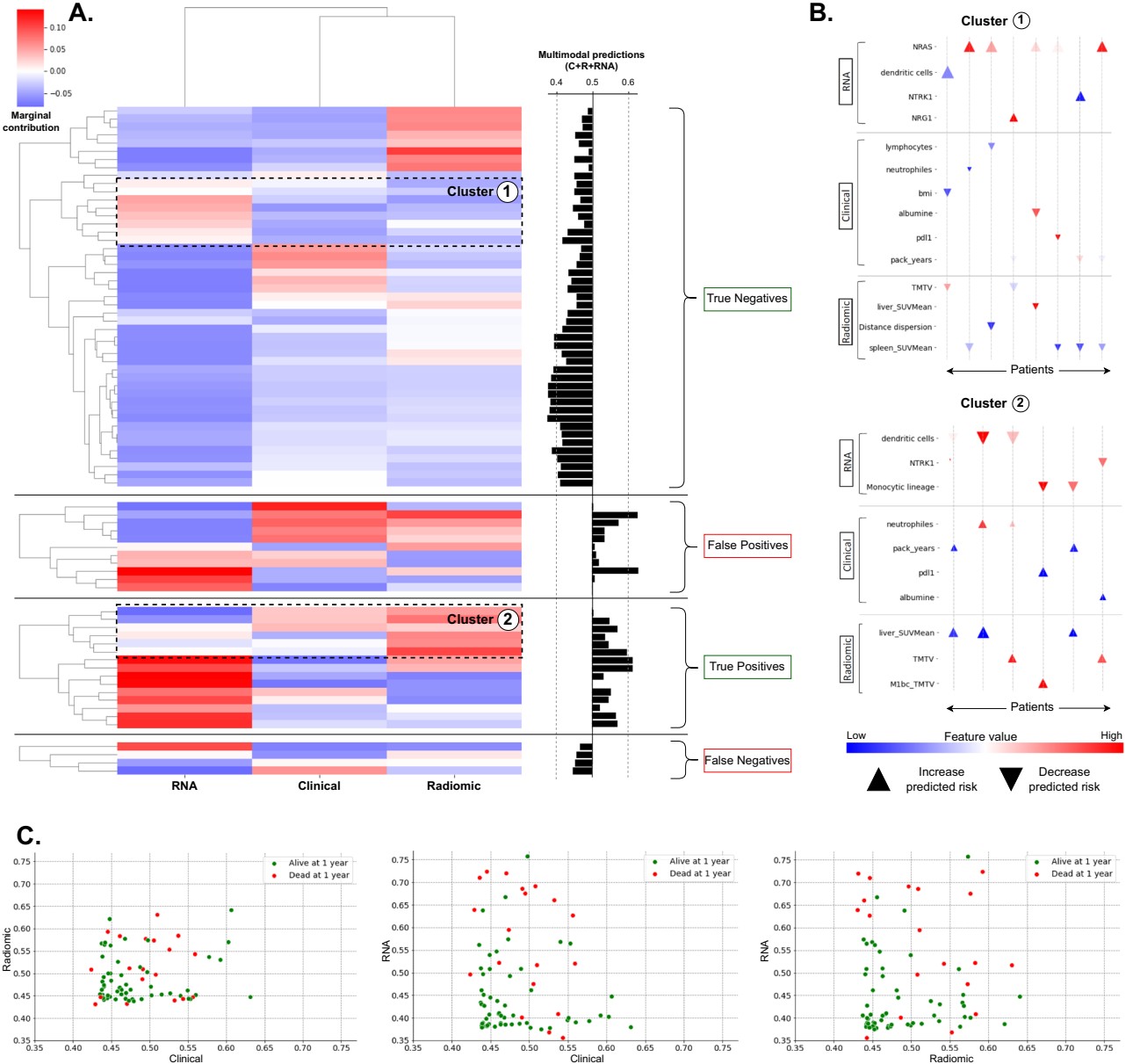

**Fig. 4 | Marginal contribution of each modality to the multimodal predictions for late fusion strategy and XGBoost classifiers. A** Heatmap of the marginal contribution (i.e., Shapley value) of each modality to the 1-year death prediction using the C + R + RNA late fusion model with XGBoost classifiers. Marginal contributions indicate how each modality influences the prediction relative to a random baseline of 0.5. Patients are stratified based on the multimodal model's final prediction (with a 0.5 threshold), where the positive class corresponds to those who died within 1 year, and the negative class corresponds to those who survived. **B** For each modality and patient in clusters 1 and 2 (see **A**), represented by vertical lines, this plot shows the feature with the highest SHAP value that aligns with the modality's marginal contribution. The size of each triangle indicates the absolute

SHAP value, while its orientation corresponds to its sign (up for positive values that increase the predicted probability of death within 1 year and down for negative values that decrease it). The color scale represents the associated feature value relative to the whole patient cohort. **C** Relationship between the unimodal predictions from clinical, radiomic, and RNA modalities (i.e., unimodal tree ensemble models). Each dot is colored according to the patient's true label. *In these plots, all marginal contributions, SHAP values, and predictions were obtained for the 77 patients with complete multimodal profiles and available 1-year death labels across the cross-validation test sets. They were collected for each of the 100 cross-validation schemes (see Methods) and subsequently averaged for each patient. Source data are provided as a Source Data file.

group membership was defined by optimizing the cutoff on the training set of each cross-validation fold to maximize the log-rank test statistic and then applying that cutoff to the corresponding test set. In the case of classification tasks, a 0.5 cutoff on the predicted probabilities was also considered (see Methods). The clinical modality alone effectively separated patients into two risk groups, with the predictions of a linear model trained to predict 1-year death yielding the best *p*-value (log-rank *p*-value = 1.26e-06, Fig. 8B). For all prediction tasks, a range of one to seven multimodal models, out of the 56 possible

models (i.e., integration strategy + multimodal combination), demonstrated superior risk stratification (as measured by the log-rank test statistic) compared to the clinical models with the lowest log-rank *p*-values (Fig. 8A and Supplementary Fig. s24). Specifically, a combination of clinical, pathomic, and RNA modalities, trained to predict 1-year death with a tree ensemble algorithm, yielded the best *p*-value (log-rank *p*-value = 3.51e-09, Fig. 8B). We further compared the multimodal score resulting from this combination (i.e., the test predictions averaged across the different cross-validation schemes) with the

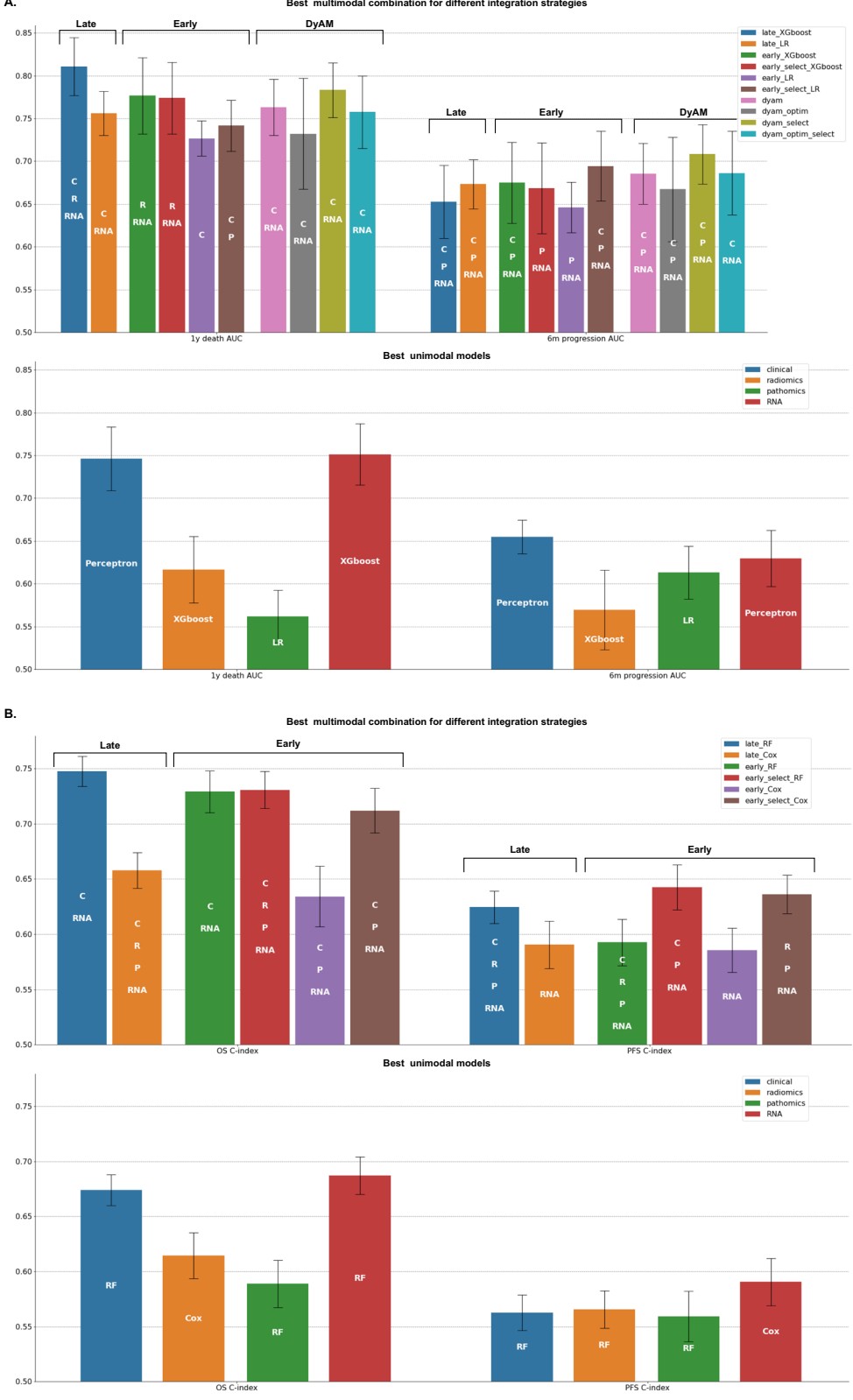

**Fig. 5 | Best unimodal and multimodal performances across all the possible combinations of modalities and predictive algorithms.** The top barplot displays the performance of the best multimodal combination for each integration strategy, while the bottom barplot shows the performance of the best unimodal algorithm for each data modality. Bar heights and error bars correspond to the mean metric (AUC or C-index) and ±1 standard deviation, respectively, estimated across the 100 cross-validation schemes (except for the dyam_optim models for which only 10 cross-validation schemes were used, due to computational constraints). **A** Best performance (AUC) for the prediction of 1-year death and 6-month progression ($n = 77$ for 1-year death and $n = 75$ for 6-month progression). **B** Best performance (C-index) for the prediction of OS and PFS ($n = 79$ for OS and $n = 80$ for PFS). Source data are provided as a Source Data file.

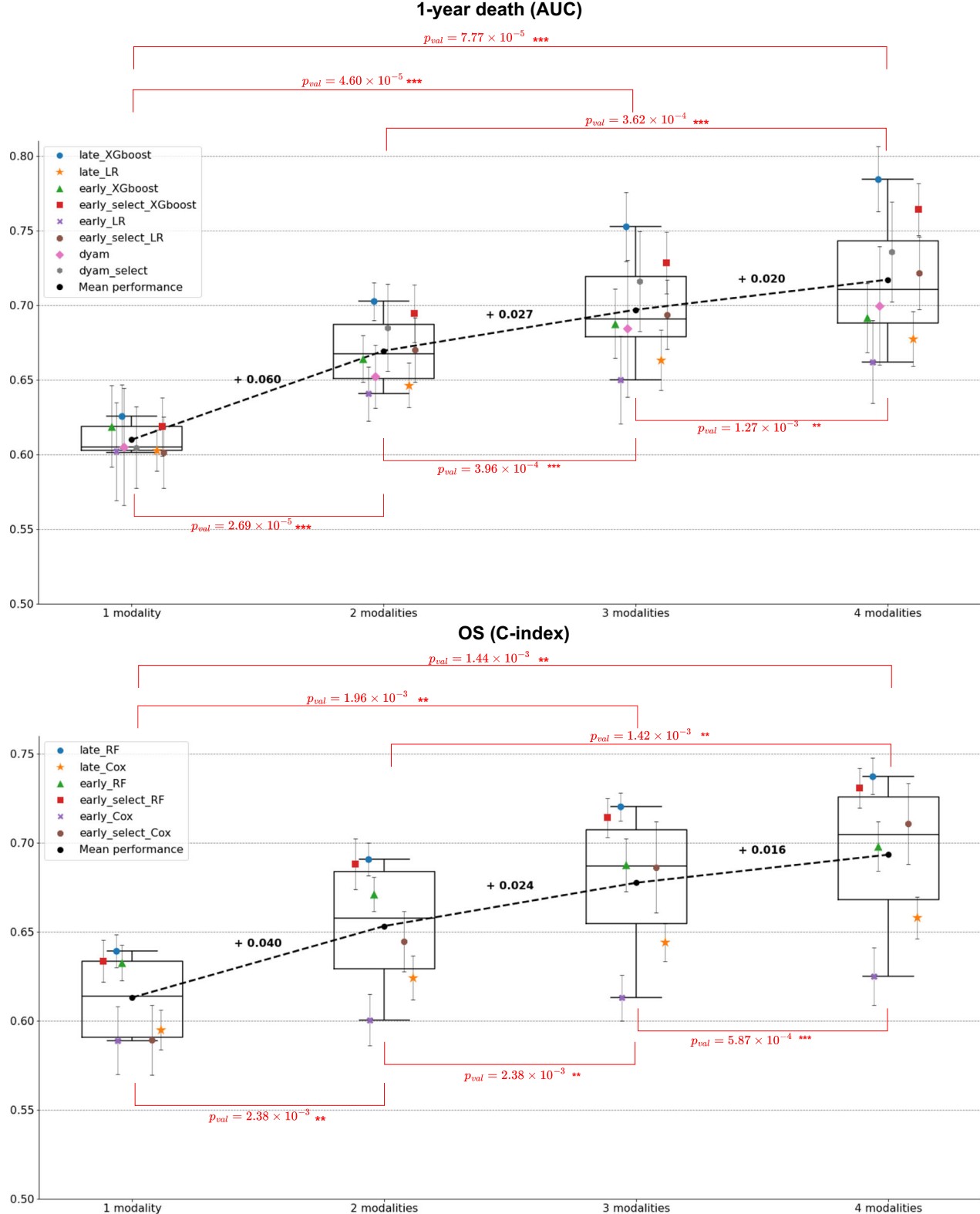

**Fig. 6 | Average performance across all models with 1, 2, 3, and 4 modalities for 1-year death and OS.** Markers and error bars correspond to the mean average performance and ±1 standard deviation respectively, estimated across the 100 cross-validation schemes. The box-and-whisker plots show the three quartiles and the minimum and maximum as whiskers up to $1.5 \times IQR(25–75\%)$. Mean increases are represented with dashed lines and bold annotations. Red annotations correspond to two-sided paired sample $t$ test $p$-values to compare the different numbers of integrated modalities (e.g., 1 modality vs 2 modalities), with $n_{models} = 8$ for 1-year death and $n_{models} = 6$ for OS. *: 1e-2 $< p_{val} \leq$ 5e-2, **: 1e-4 $< p_{val} \leq$ 1e-3, ***: $p_{val} \leq$ 1e-4. Source data are provided as a Source Data file.

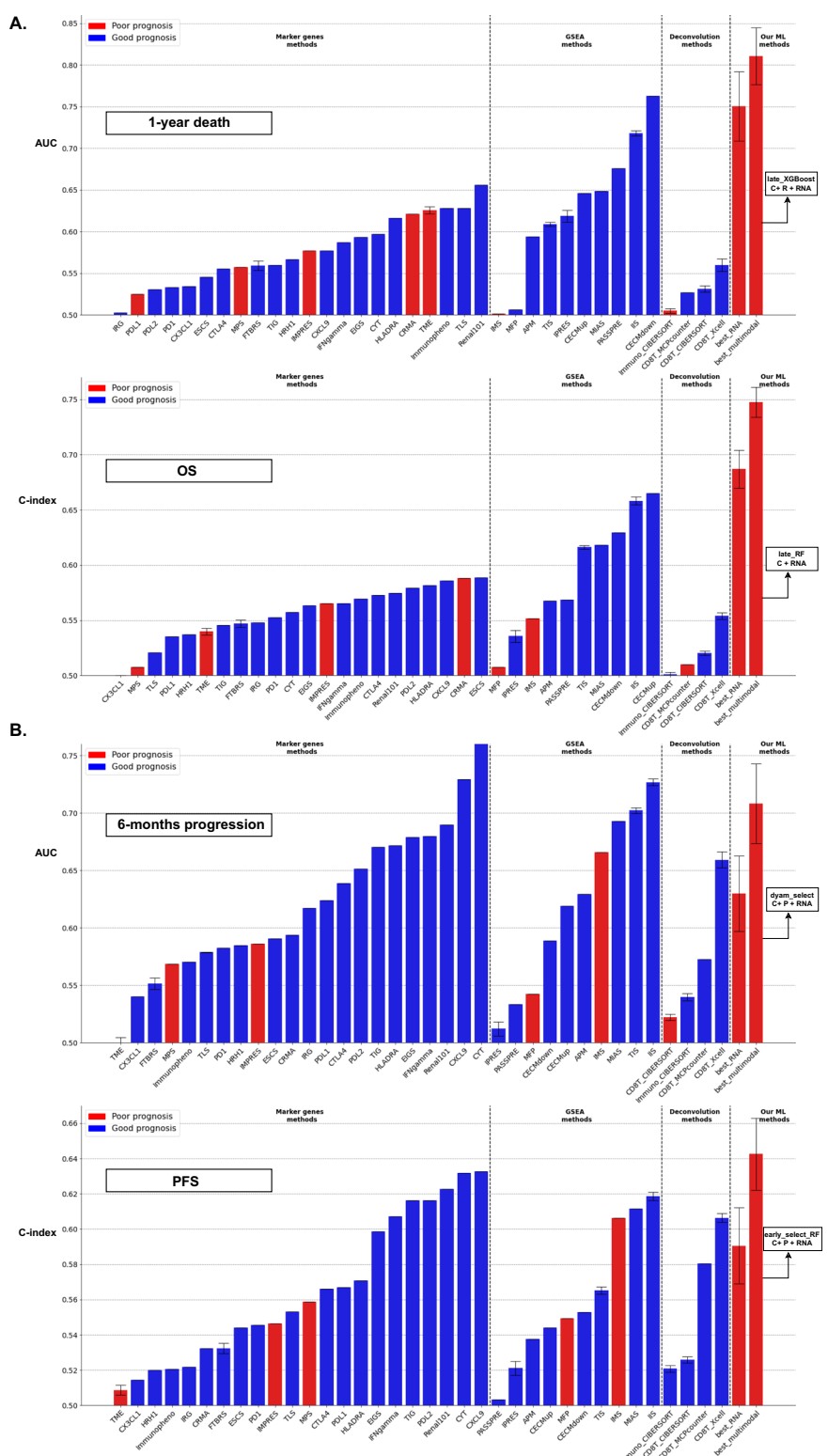

**Fig. 7 | Comparison of the performance of transcriptomic signatures with our best transcriptomic and multimodal models.** Comparison of the performance of 36 transcriptomic signatures previously associated with immunotherapy (from the literature) against the best unimodal transcriptomic model and the best multi-modal model from our analysis for each prediction task. The bar height corresponds to the performance metric (either ROC AUC or C-index), averaged across 100 cross-validation schemes and estimated for the 80 patients with a complete multimodal profile. The error bar indicates ±1 standard deviation (for signatures without a training step, this standard deviation is zero). Performance metrics were transformed using max(x, 1-x) to account for signatures with a performance below 0.5. Blue bars represent performances below 0.5 (higher signature values are associated with better prognosis), while red bars represent performances above 0.5 (higher signature values are associated with worse prognosis). **A** Comparison for 1-year death prediction (*n* = 77 patients with a complete profile and available 1-year death label) and OS prediction (*n* = 79 patients with complete profile and available OS information). **B** Comparison for 6-month progression prediction (*n* = 75 patients with complete profile and available 6-month progression label) and PFS prediction (*n* = 80 patients with complete profile and available PFS information). Source data are provided as a Source Data file.

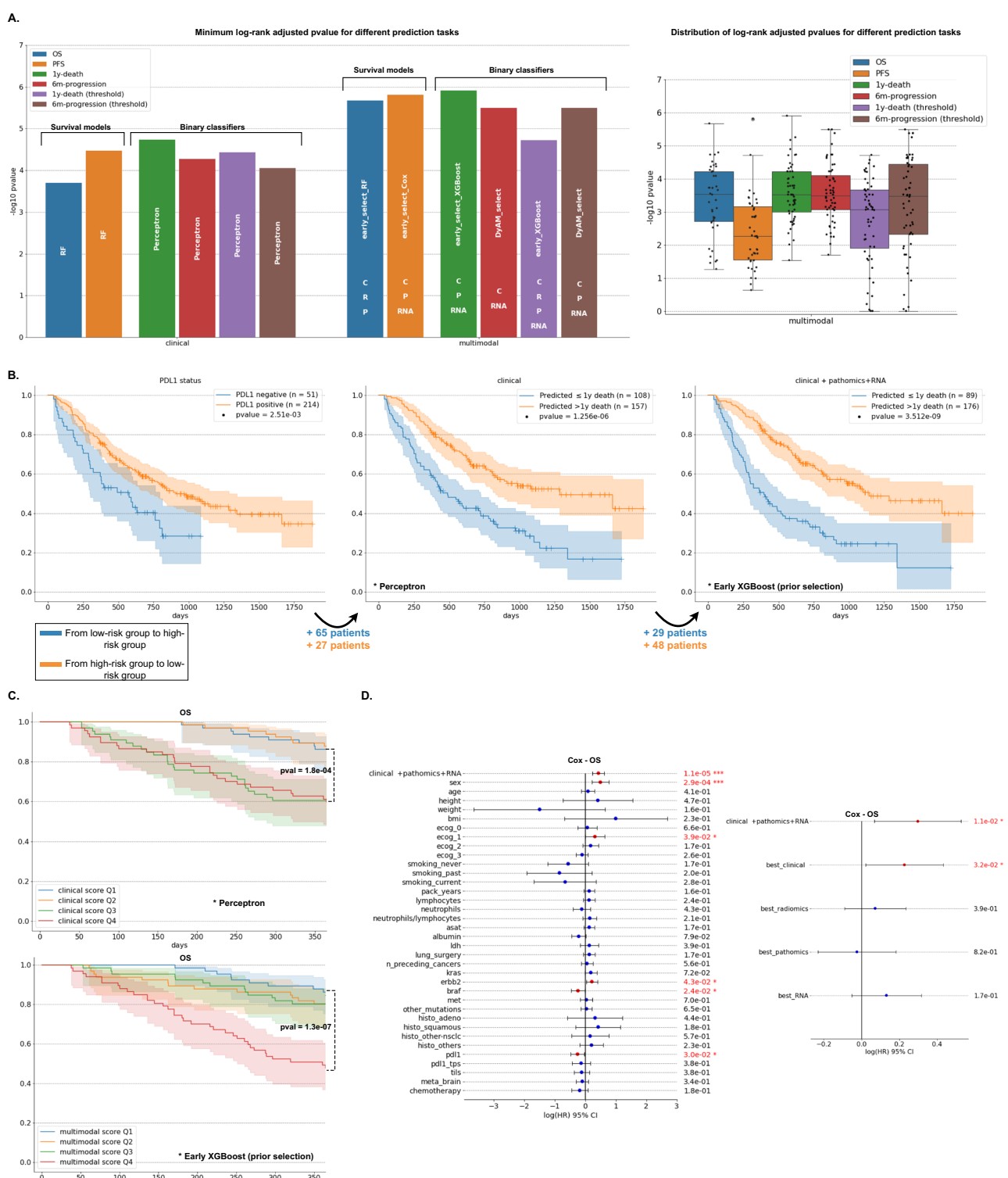

clinical score derived from the linear model described above (Fig. 8B). To do so, we divided the cohort into quartiles based on these two scores and performed Kaplan-Meier analysis for OS within the first year of therapy (Fig. 8C). Both scores yielded a lowest quartile group (low risk) with a 14% death rate (9/66 patients) within the first year of therapy. However, the multimodal score identified a highest quartile group (high risk) with a 52% death rate (35/67 patients, 20 treated with immunotherapy + chemotherapy and 15 treated with immunotherapy only), whereas the clinical score yielded a highest quartile group with a

40% death rate (27/67 patients, 13 treated with immunotherapy + chemotherapy and 14 treated with immunotherapy only).

Finally, the multimodal score resulting from the combination of clinical, pathomic, and RNA modalities demonstrated a significant association with OS when integrated into a multivariate Cox model along with the clinical features (Fig. 8D). Likelihood-ratio tests indicated a significant effect of this multimodal score compared to a Cox model fitted only with clinical information collected from routine care (p-value = 1.09e-05). Five clinical variables were also significant,

**Fig. 8 | Risk stratification and survival analysis for OS with the predicted multimodal scores.** **A** Comparison of the stratification of the patients into high-risk and low-risk groups for OS, for different predictive tasks, with log-rank *p*-values (*n* = 265 patients with the 4 targets available for a fair comparison). Only the combinations, including the clinical modality, are compared (see Methods). On the left, clinical and multimodal models are compared by showing the lowest log-rank adjusted *p*-values from all clinical (left) and multimodal (right) models for each prediction task. On the right, the box-and-whisker plots show the three quartiles, with whiskers extending up to 1.5 × *IQR* (25–75%) to show the range of adjusted *p*-values. **B** Kaplan-Meier survival curves (solid lines) with 95% confidence interval (shaded areas) for the high-risk and low-risk OS groups defined by PDL1-status (left), the clinical model with the lowest log-rank *p*-value (middle), and the multimodal model with lowest log-rank *p*-value (right). Unlike in **A**, unadjusted *p*-values

are displayed here. **C** Kaplan-Meier survival curves (solid lines) with 95% confidence interval (shaded areas) for OS within the first year of therapy. The cohort is stratified into quartiles based on either the clinical score derived from the clinical perceptron predictions (top) or the multimodal score derived from the predictions of the clinical + pathomics + RNA model (bottom). **D** Log hazard ratios (points) with 95% confidence intervals (error bars) and likelihood-ratio test *p*-values associated with multivariate Cox models trained to predict patient's OS (*n* = 265). Cox model with the clinical + pathomics + RNA score as well as the clinical features collected in this study (left). Cox model with the clinical + pathomics + RNA score, as well as the best unimodal scores derived from the top performing unimodal models for 1-year death prediction (right), with the best clinical score corresponding to the clinical perceptron identified in panels (**B** and **C**). Source data are provided as a Source Data file.

including sex, ecog_1, braf, errb2, and pdl1 (see Supplementary Methods). When comparing the multimodal score with the best unimodal scores (i.e., derived from the top-performing models for 1-year death prediction, Fig. 5A), only the multimodal score (hazard ratio (HR) = 1.35, 95% CI [1.07-1.69], *p*-value = 1.11e-02) and the clinical score (HR = 1.25, 95% CI [1.02-1.54], *p*-value = 3.17e-02) demonstrated a significant effect (Fig. 8D).

## Discussion

Multimodal approaches for developing accurate biomarkers for the outcome of metastatic NSCLC patients treated by immunotherapy are highly promising but have been rarely explored so far. In this study, we built a new multimodal NSCLC cohort to investigate the benefit of integrative strategies. We extracted interpretable features from clinical data, PET/CT images, digitized pathological slides, and bulk RNA-seq profiles and compared the performance of unimodal and multimodal machine learning models to accurately predict patient outcomes. We conducted an extensive exploration of different algorithms, integration strategies, and outcome encodings (i.e., binary vs. continuous targets) to highlight consistent trends that remain robust regardless of the specific choices made within the analysis pipeline.

We trained several unimodal models capable of predicting OS, 1-year death, PFS, and 6-month progression using pre-treatment clinical and transcriptomic data. Our design choice to base the entire workflow on interpretable features enabled us to conduct a thorough feature importance analysis. It revealed that clinical models integrated previously established biomarkers into efficient multivariate predictive models[7], while RNA models used signatures of the Tumor Micro-Environment (TME)[20] as well as the expression of specific oncogenes, which were robust to the biopsy location. Notably, our analysis highlighted the positive impact of a high abundance of dendritic cells (DC), as scored by MCP counter[20], on patient survival. These findings thus provide further evidence of the potential of RNA-seq data to predict immunotherapy response. They corroborate recent studies that have demonstrated the enhanced predictive power of RNA-seq data compared to conventional modalities used in clinical practice, such as mutational data or immunohistochemistry[21,22]. Radiomics and pathomics, used as standalone predictors, exhibited limited predictive ability in our analysis. Radiomic models predominantly relied on the TMTV to predict OS and 1-year death, confirming its strong predictive value[23]. Nonetheless, the other aggregated features that we investigated did not increase performance, highlighting the need to explore and design additional radiomic features that could effectively complement TMTV.

The importance of DCs for the prediction of patient outcomes is in line with previous pre-clinical mechanistic studies that highlighted their central role in shaping anti-tumor immunity[24,25]. In particular, type 1 DCs (DC1) present antigens to CD8 + T cells, and their abundance in the TME has been linked to increased survival and improved response to immunotherapy in both animal models[26] and human

cancer lesions[27]. Tertiary lymphoid structures (TLS) are ectopic formations containing high densities of B cells, T cells, and dendritic cells, at sites of persistent inflammatory stimulation, including tumors[28]. Given the accumulating evidence linking the presence of tumor TLS and good prognosis in cancer patients[29], we investigated a possible association between DCs and TLS. However, visualization of H&E sections and the poor correlation between B cells and DCs in our data did not support this hypothesis. The link between DCs and prognosis/survival observed in the present study can most likely be explained by DCs' ability to capture tumor antigen in the tumor lesion and present it to T cells in draining lymph nodes.

Our study provided further evidence supporting the superiority of multimodal over unimodal approaches to build accurate biomarkers for the outcome of metastatic NSCLC patients treated with immunotherapy. In all prediction tasks related to OS, 1-year death, PFS, and 6-month progression, we identified multiple multimodal combinations that outperformed the unimodal models, including the models relying on standard clinical data. Furthermore, several combinations demonstrated enhanced patient risk stratification for OS, outperforming the best clinical model across the whole cohort. Multivariate Cox models, combined with Kaplan-Meier analyses, underscored the enhanced prognostic value provided by multimodal predictions beyond routine clinical biomarkers. They further highlighted that multimodal scores could help better identify patients with the most severe prognosis, thereby guiding tailored treatment strategies such as intensified follow-up care or considering chemotherapy even in cases with high PD-L1 expression. No single integration strategy outperformed others for all prediction tasks, but most of them effectively built multimodal predictors that outperformed the best unimodal predictors. While we confirmed the potential of the DyAM method[16], especially with prior feature selection, we found that a much simpler model based on late fusion frequently compared favorably to more complex integration strategies. We assume that the robust performance of the simple late fusion approach is due to its ability to handle missing modalities. Although it is not ruled out that more complex methods might ultimately yield better results on ideal and large datasets, it should be considered that such ideal scenarios will be quite rare in clinical reality. Overall, our comprehensive benchmark of multiple algorithms and integration strategies highlighted a consistent benefit of combining multiple modalities to predict immunotherapy outcomes, irrespective of specific settings or methodologies.

Our study has several limitations. First, we dealt with a relatively modest number of patients, many of whom had missing data, which limited the statistical power of our analyses, particularly when testing the association between patient survival under immunotherapy and standard clinical biomarkers (Fig. 1C and Supplementary Fig. S3). Besides, we did not have access to an external validation cohort to assess the reproducibility and robustness of our results. Due to missing modalities, our evaluation was conducted on a subset of 80 patients with a complete multimodal profile to ensure a fair comparison between multimodal combinations. Therefore, the absolute

performance scores highlighted in this study should be interpreted cautiously. Nevertheless, the relative comparison of these scores between different combinations of modalities, as well as in comparison with random predictors and standard clinical biomarkers, all trained and evaluated with the same methodology, remains valuable. It confirms the superiority of multimodal approaches over unimodal models and thus may motivate the collection of new large and multi-centric multimodal NSCLC cohorts. These multi-centric cohorts are needed to address challenges related to missing modalities, particularly RNA, and further validate our findings. Furthermore, our study did not include advanced NSCLC patients who did not receive immunotherapy, which prevented us from distinguishing between predictive and prognostic biomarkers[30]. It is likely that some features highlighted in our analyses, particularly clinical and radiomic ones, are more prognostic than predictive, as they have already been associated with patient outcomes for other treatments[31,32]. Nonetheless, our study identified promising features, especially transcriptomic ones, as well as several combinations of features −whether multimodal or not−that foster further research to evaluate their predictive value and their association with immunotherapy response. In addition, our data collection pipeline involved time-consuming processes, such as the manual segmentation and annotation of PET/CT scans by nuclear medicine physicians, which could deter the collection of new external cohorts and limit their size. However, deep learning methods for automatically segmenting lesions on PET scans[33] or computing surrogate radiomic features[34] have recently shown very promising results and may soon be incorporated into the multimodal pipeline to overcome this bottleneck. Finally, despite being one of the most powerful modalities in our analysis, the RNA modality is not yet routinely available in clinical practice in many places, unlike clinical information, PET/CT images, or pathological slides. Its collection involves additional costs and is often affected by the low quality of the remaining tissue samples from the biopsy. However, it could be used as an initial step to identify prognostic mechanisms, which could subsequently be assessed with more cost-effective technologies.

Our multimodal cohort allowed us to demonstrate the ability of clinical, radiological, pathological, and transcriptomic data to inform powerful multimodal predictors for the outcome of patients treated with immunotherapy in metastatic NSCLC. It provided several promising predictors that outperformed both established biomarkers (e.g., PD-L1 expression) and unimodal predictors. They now require refinement and validation in multicentric studies. These results foster further efforts to gather large multimodal cohorts and explore multimodal biomarkers that could efficiently guide therapeutic decisions.

## Methods

This study was approved by the institutional review board at Institut Curie (DATA200053) and informed consent from all patients was obtained through institutional processes. Data were de-identified, collected, and stored in compliance with GDPR.

### Clinical data

Baseline clinical data for all 317 patients were collected from the Electronic Medical Record of Institut Curie Hospital using a predefined case report form based on the ESME-AMLC database.

Each patient's response to immunotherapy was assessed through OS and PFS. OS was defined as the duration from the initiation of first-line immunotherapy (with or without chemotherapy) to the patient's death or last available status update. PFS was defined as the duration from the initiation of first-line immunotherapy to the occurrence of the first progression event or last available status update, including the emergence of new lesions or the progression of pre-existing ones. We also considered binary outcomes, specifically 1-year death (0 for patients who were still alive after one year of immunotherapy and 1 otherwise) and 6-month progression (0 for patients

whose disease did not progress after six months of immunotherapy and 1 otherwise). Patients whose OS or PFS was censored before one year or six months, respectively, were excluded from the analysis with binary outcomes.

We selected 30 baseline clinical features for our predictive models. A detailed list of these features and their definitions are provided in Supplementary Methods. We applied one-hot encoding to all categorical features.

### Radiomic data

Baseline [18 F]FDG-PET /CT scans were collected for 201 patients. Two experienced nuclear medicine physicians delineated all tumor foci in all PET scans using LIFEx software v.7.3 (https://www.lifexsoft.org/)[35]. In addition, they annotated the location of each lesion using an anatomical partition inspired by TNM staging[36]. For instance, ipsilateral and contralateral lung metastases were distinguished since they are not associated with the same TNM stage. Subsequently, all images were resampled to a fixed 2x2x2 mm3 voxel size, and the segmented tumor regions were processed by applying a fixed threshold of 2.5 standardized uptake value (SUV) units to exclude voxels with SUV values below this threshold. The SUVmax value, the volume, and the centroid of each resulting tumor region were extracted with the IBSI-compliant PyRadiomics Python package[37,38]. The SUVmean values from spherical ROIs manually delineated in the healthy regions of the liver and spleen were also extracted for each patient, without prior 2.5 SUV thresholding (liver ROIs mean volume = 24 cm$^3$, standard deviation 18 cm$^3$ – spleen ROIs mean volume = 10 cm$^3$, standard deviation 8 cm$^3$).

All these extracted data were then aggregated into 30 baseline whole-body radiomic features to capture the spread of the metastatic disease as well as its metabolic activity. A detailed list of these features and their definitions is provided in Supplementary Methods. We considered the SUVmean values of healthy ROIs in the spleen and the liver, the TMTV[23], and the number of invaded organs visible on the PET scan, including the lungs, sub- and supra-diaphragmatic lymph nodes, the pleura, the liver, the bones, the adrenal gland, and a final category for other regions. In addition, using the centroids of the processed tumor regions, we calculated the standardized Dmax[39] – the largest distance between two lesions normalized by the body surface area - and the quartile dispersion of the distances between each tumor region's centroid and the global centroid. Finally, for each TNM stage, we took into account all the tumors located in associated regions and computed the TMTV as well as the mean, standard deviation, and maximum value of all the SUVmax values. For instance, for the T stage, we considered the primary lung tumor as well as all the ipsilateral lung metastases. In cases where no tumor was present in these regions, the feature values were set to zero. Except for features associated with SUVmax, we excluded lesions corresponding to lymphangitic spread, diffuse pleural metastases, diffuse myocardial metastases, and diffuse subdiaphragmatic metastases because their accurate segmentation was questionable.

All the features associated with metabolic volume were log-transformed (i.e., $\log(x+1)$) to deal with right-skewed distributions.

### Pathomic data

Baseline pathological slides stained with Hematoxylin-Erythrosine-Saffron (HES) were collected from the FFPE biopsy blocks of 236 patients and subsequently scanned by the Experimental Pathology Platform at Institut Curie. From each slide, we segmented all nuclei with a custom automatic pipeline derived from Lerousseau et al.[40], trained in a weakly supervised setting with publicly available data from TCGA as well as data sets from Institut Curie. The slides from the current study were not used to train the segmentation pipeline.

The cell nuclei of six cell types were annotated on each slide, including stromal, epithelial, dead, tumor, connective, and inflammatory cells. These annotations were then used in a pathomic approach[41]

to extract 134 relevant features, characterizing the density, the relative proportion, or the spatial organization of these different cell types within the scanned tissue.

## Transcriptomic data

Residual FFPE biopsy specimens containing sufficient RNA were collected for 134 patients and RNA sequencing was performed at the Sequencing Core facility of Institut Curie with the Illumina TruSeq RNA Access technology. The RNA-seq data were then processed with the Institut Curie RNA-seq pipeline v4.0.0[42]. The raw bulk RNA-seq read counts were normalized with TPM (Transcripts Per Million) and log-transformed (i.e., $\log(x+1)$).

The abundance of 8 immune cells and 2 stromal cells in the Tumor Micro-Environment was estimated using the MCP-counter method[20]. In addition, log expressions of 22 oncogenes associated with lung cancer were used as features (*KRAS*, *NRAS*, *EGFR*, *MET*, *BRAF*, *ROS1*, *ALK*, *ERBB2*, *ERBB4*, *FGFR1*, *FGFR2*, *FGFR3*, *NTRK1*, *NTRK2*, *NTRK3*, *LTK*, *RET*, *RIT1*, *MAP2K1*, *DDR2*, *ALK*, and *CD274*). The biopsy site was also considered as a categorical feature and one-hot encoded, distinguishing between lungs, pleura, lymph nodes, bones, liver, adrenal gland, and brain. This information was available for 84 patients. Finally, we used a custom pipeline derived from Jessen et al.[43] to estimate the Tumor Mutational Burden (TMB) from the RNA-seq reads mapped to the reference genome hg19 with STAR aligner (1-pass). VarDict tool[44] was used for variant calling and several filters were applied to detect somatic variants. This feature, named TMB_RNA, was available for 110 patients.

## Genomic data

TMB was estimated for 43 patients using a custom NGS panel of 571 genes called DRAGON (Detection of Relevant Alterations in Genes involved in Oncogenetics by NGS) and marketed by Agilent under the name of SureSelect CD Curie CGP. Only non-synonymous alterations (excluding splice site) were considered, with a threshold of 15 mutations per megabase used to distinguish between patients with high and low TMB.

## Unimodal analyses - Tree ensemble methods

The Extreme Gradient-Boosted Trees algorithm[19] implemented in XGBoost v.1.7.6 Python package was used to solve 1-year death and 6-month progression classification tasks. We used default parameter values except for *scale_pos_weight* which controls the balance between positive and negative weights and was set to the proportion of negative over positive labels estimated with the training set. Each XGBoost classifier was calibrated with Platt's logistic model[45]: predictions were collected with a 10-fold stratified cross-validation scheme on the training set and then used as input for a univariate logistic regression model with balanced class weights. To generate the final calibrated predictions, this logistic model was applied to the raw predictions of the XGBoost classifier.

The Random Survival Forest algorithm[18] implemented in scikit-survival v.0.21.0 Python package was used to solve survival tasks for OS and PFS. We used default parameter values except for *max_depth* which controls the size of the survival trees and was set to 6 to mitigate the risk of overfitting. Contrary to XGBoost, Random Survival Forest algorithm does not handle missing values automatically. Therefore, we applied median imputation for continuous features and most-frequent imputation for categorical features, both fitted to the training set.

## Unimodal analyses - linear methods

The Logistic Regression algorithm with elastic net penalty implemented in scikit-learn v.1.2.2 Python package was used to solve 1-year death and 6-month progression classification tasks. We used the saga optimizer, a regularization parameter $C = 0.1$, an L1 ratio of 0.5, a maximum number of iterations of 2500, and balanced class weights.

The Cox's proportional hazard's algorithm with elastic net penalty implemented in scikit-survival v.0.21.0 Python package was used to solve survival tasks for OS and PFS. We used default parameter values except for *alpha_min_ratio* which controls the regularization strength and was set to 0.01.

In both algorithms, we preprocessed the data by first applying robust scaling, followed by median imputation for continuous features and the most frequent imputation for categorical features. All these operations were fitted to the training set.

## Multimodal analyses - late fusion

We used a late fusion strategy to combine every possible subset of modalities. This analysis was limited to fusions of the same predictive algorithms with the same parameter values. For instance, for classification tasks, we separately explored the fusion of penalized logistic regression models and the fusion of XGBoost models.

First, we restricted the training set to patients with at least one of the modalities of the combination of interest available. Then, we independently trained each unimodal model using the subset of patients in the training set for whom the associated modality was available. Finally, for each patient in the test set, we computed the multimodal prediction by averaging the unimodal predictions for the available modalities. For survival models, the unimodal predictions were standardized before averaging, using the mean and standard deviation values estimated for each modality based on predictions obtained from a 10-fold cross-validation scheme applied to the training set (stratified with respect to the censorship rate). XGBoost late fusion models were also calibrated using Platt's logistic model[45], following the strategy described previously.

## Multimodal analyses - early fusion

We used an early fusion strategy to combine every possible subset of data modalities and compared the results with those obtained with the late fusion strategy. The training set was once again limited to patients with at least one available modality. We first pre-processed each data modality separately, considering the subset of patients within the training set for whom that modality was available. Subsequently, we concatenated the processed features from all the modalities to form the input for the predictive model. We used the same models and the same parameter values as in the unimodal analyses (i.e., XGBoost, Logistic Regression, Random Survival Forest, and Cox model). XGBoost early fusion models were calibrated using Platt's logistic model[45].

For linear models, missing modalities were handled by replacing them with zero values. XGBoost did not require special handling for missing modalities, while for Random Survival Forest, a double-coding strategy was applied, inspired by Engemman et al.[46]. This approach involved duplicating features, assigning either very high values or very low values to patients with missing modalities. This allowed the survival tree to decide on which side of the decision split to place patients with missing modalities.

We also explored early fusion with a preliminary feature selection step to maintain a consistent number of features across different multimodal combinations. We first calculated a univariate score for each feature using $|m - 0.5|$, where $m$ corresponds to either the AUC or the C-index computed with the training set. We then ranked all the features from all the modalities accordingly. To reduce redundancy, we filtered out highly correlated features by iterating through the ranked feature list from top to bottom and removing subsequent features with a Pearson correlation exceeding $\rho = 0.7$. Finally, for each modality, we selected the top $\lfloor n_{total}/n_{modas} \rfloor$ features from the filtered, ranked list that belonged to that modality, where $n_{total}$ corresponds to the number of features to keep in the multimodal model (in this analysis, we used $n_{total} = 40$) and $n_{modas}$ corresponds to the number of

modalities in the multimodal combination of interest. For unimodal combinations, this feature selection step was ignored.

## Multimodal analyses - DyAM model

We used our own implementation of the DyAM model, with PyTorch v.2.0.1 Python package, to combine every possible subset of data modalities and compared the results with those obtained with late fusion and early fusion strategies. We adopted the exact same architecture as described in Vanguri et al.[16], which used single-layer feedforward neural networks with a tanh activation function for unimodal predictions and single-layer feedforward neural networks with a softplus activation function for unimodal attention weights. Furthermore, similar to ref. 16, we trained our models with a binary cross-entropy loss with balanced class weights, a learning rate of 0.01, 125 training epochs, an L2 regularization strength of 0.001, and the Adam optimizer.

Data were pre-processed with robust scaling as well as median imputation for continuous features and most-frequent imputation for categorical features. For pathomic data, we also applied a Principal Component Analysis (PCA) step with 40 components to reduce the size of the neural networks. We also applied a preliminary feature selection step as described previously.

We implemented a nested cross-validation scheme with inner 10-fold stratified cross-validation and a grid-search strategy to optimize the learning rate and the L2 regularization strength for each combination. Due to computational constraints, we limited the number of repetitions to 10 in these cases.

## Statistical analysis – performance evaluation

All the predictive models were trained and tested using a 10-fold cross-validation scheme applied to the entire cohort. The folds were stratified based on class proportion for classification tasks and censorship rate for survival tasks. In each fold and for each modality combination, patients with all modalities missing were excluded from the training set. Pre-processing operations, including missing value imputation, scaling, or univariate feature selection, were fitted to each training set and then applied to the corresponding test set to prevent any data leakage (Supplementary Fig. s4). This process was repeated 100 times, with data shuffling in each iteration. We used the same repeats across all the experiments.

The performance of each model was evaluated using Uno's Concordance Index (C-index)[47] for survival tasks and the Area Under the ROC Curve (AUC) for classification tasks. These metrics were computed for each cross-validation scheme, considering only the predictions in the test sets of the 80 patients with a complete multimodal profile to ensure a fair comparison among the different combinations (Supplementary Fig. s4). Subsequently, the metrics were averaged over all 100 repetitions, and their standard deviation was calculated to measure the variability resulting from the random partition of the data into 10 folds.

## Statistical analyses - permutation tests

The significance of the results was assessed with one-sided permutation tests, running the pipeline described above 100 times with randomly shuffled outcomes. Permutation tests were also used for univariate predictors, along with 10,000-repeated bootstrap sampling for computing 95% confidence intervals.

To compare the performance of the different multimodal models and test for statistically significant differences, we applied a two-step procedure. First, for each pair of combinations $(i, j)$ and each cross-validation scheme $s$, the superiority of $j$ over $i$ was assessed with a one-sided paired permutation test[48], resulting in 100 $p$-values $\left(p_{ij}^s\right)_{s=1}^{100}$. Subsequently, for each pair of combinations $(i, j)$, these 100 $p$-values were adjusted using the Benjamini-Hochberg procedure (FDR controlled at level $\alpha = 0.05$), and the frequency of statistically significant

tests across the 100 tests was computed. The performance of the different predictive models was estimated on the subset of patients with a complete multimodal profile. Although the paired permutation test described in Bandos et al.[48] was originally designed to compare two AUCs, we extended it to the comparison of two C-indexes since both metrics are C statistics and the C-index can be seen as a generalization of the AUC for censored survival data.

## Survival analyses

For overall survival, we evaluated the ability of each predictive model to stratify patients into two distinct risk groups, including the models trained to predict PFS or 6-month progression. First, for each fold of each cross-validation scheme, we explored a range of thresholds going from the 30th to the 70th percentiles of the training predictions and selected the one that minimized the log-rank $p$-value on the training set. For PFS-related models, we focused on the stratification of PFS to find the best threshold, mimicking scenarios where OS is not available during the training process. For classification tasks, the 0.5 threshold was also considered. The learned threshold was then applied to the corresponding test set, assigning patients to a low-risk group or a high-risk group for overall survival. Risk group membership was thus collected for each patient across the test sets of the cross-validation scheme. This resulted in 100 group memberships for each patient, corresponding to the 100 cross-validation schemes. Finally, these 100 assignments were aggregated by calculating the frequency of low-risk and high-risk group assignments for each patient. Patients with a frequency of high-risk group greater than 50% were assigned to the final high-risk group, while those with a frequency strictly lower than 50% were assigned to the final low-risk group. We compared the survival distributions of the final low- and high-risk groups for each predictive model with Kaplan-Meier curves and a log-rank test. The Benjamini-Hochberg procedure was used to control for multiple testing (FDR controlled at the level $\alpha = 0.05$). We focused on the subset of multimodal combinations which included clinical data to work with a sufficiently large cohort (i.e., 265 patients with the 4 targets available for fair comparisons). This analysis thus assessed the risk stratification ability of multimodal models that incorporated multiple modalities alongside clinical data, whenever they were available.

Finally, we derived a score from each multimodal predictive model by collecting its predictions from the test sets of each cross-validation scheme and averaging them over 100 repeats. This score was used as input to a multivariate Cox model to predict patients' OS, along with clinical features or unimodal scores. The Cox model was then fitted on the 265 patients with the 4 targets available (i.e., OS, 1-year death, PFS, and 6-month progression) for fair comparison between the models. All the input variables were first standardized to ensure comparable hazard ratios. To address missing clinical values, we used median imputation for continuous clinical features and the most frequent imputation for categorical clinical features. For unimodal scores, missing values were replaced by 0.5 for classification models and 0 for survival models, since all the models were calibrated with nested cross-validation. We used lifelines v.0.27.4 Python package to fit the Cox models. Likelihood-ratio tests were computed manually with the difference between the log-likelihood of the two compared models and a chi-squared test.

## Feature importance analysis

For each algorithm and modality, we used the permutation explainer provided by SHAP v.0.42.1 Python package[49] to compute the SHAP values for each feature and each patient. SHAP values were computed only when the patient was in a test set of the cross-validation scheme. This resulted in $n_{patients} \times n_{features}$ SHAP values for each cross-validation scheme, where $n_{patients}$ corresponds to the number of patients with the modality of interest available and $n_{features}$ corresponds to the number of features extracted for this modality. All these

values were subsequently averaged across the 100 cross-validation schemes to produce the final set of $n_{patients} \times n_{features}$ mean SHAP values.

A positive SHAP value for a patient $p$ and a feature $f$ means that considering the feature $f$ in the predictive model of interest increases the patient $p$'s probability of death or progression for classification tasks and the patient $p$'s risk of death or progression for survival tasks. Conversely, a negative SHAP value means that $f$ decreases the patient $p$'s probability or risk of death or progression.

For each data modality, we applied a three steps procedure to combine the SHAP values from both related tasks (i.e., OS and 1-year death, or PFS and 6-month progression) and both approaches (i.e., linear and tree ensemble methods) and obtain a consensus ranking of features with respect to their importance in the predictive models for overall survival or progression-free survival. First, we filtered out non-robust features whose impact on the predictions was not consistent across the four predictive models (for radiomics we did not consider the Logistic Regression model since its AUC was lower than 0.5). To do so, we computed, for each feature and each model, the Spearman correlation between the SHAP values and the values of the feature of interest and then filtered out the features whose correlation sign was not consistent across the four models. Then we ranked the remaining robust features for each of the four models with respect to their absolute SHAP values averaged across all the patients. For each model $i \in \{1, \ldots, 4\}$ and each feature $f$ we thus obtained a rank $r_f^i \in \{1, \ldots, n_F\}$ (with $n_F$ the number of robust features) with 1 corresponding to the least important feature and $n_F$ to the most important one. Finally, we aggregated all these ranks across the four models to obtain a consensus ranking, taking into account the performance of each model. The consensus rank of the feature $f$ was defined as:

$$r_f^{cons} = \frac{1}{s_1 + s_2 + s_3 + s_4} \sum_{i=1}^{4} s_i r_i^f \qquad (1)$$

Where $s_i$ corresponds to the score of the model $i$ and is equal to $\max(0, score_i - 0.5)$ (the score is either the AUC or the C-index). The remaining robust features were also tested with univariate permutation tests both for OS and 1-year death (or PFS and 6-month progression): 1000 univariate AUCs or C-indexes were generated with permuted labels and then compared to the original AUC or C-index. Features that remained statistically significant after the Benjamini-Hochberg correction (FDR controlled at the level $\alpha = 0.05$) were reported in the consensus ranking.

The consensus ranks were normalized with respect to the total number of consensus features. Each rank was also assigned a sign that corresponded to the sign of the Spearman correlation coefficient between the SHAP values and the values of the associated feature. A positive sign means that the effect of the feature on the predicted risk/probability of the event increases with the feature value, while a negative sign means that the effect decreases with the feature value. In this context, the term effect is linked to the SHAP value and can be either positive (increasing predicted risk) or negative (decreasing predicted risk).

### Benchmark of transcriptomic signatures
We identified 36 transcriptomic biomarkers associated with immunotherapy response from a systematic literature search and curation[50], encompassing various cancer types and immune checkpoint inhibitors (Supplementary Table s2). They were categorized into three groups: marker genes biomarkers that focused on a subset of genes to compute an overall score for each patient (22 biomarkers), GSEA biomarkers that applied single-sample gene set enrichment analysis (ssGSEA) to compare sets of marker genes with non-marker genes (10 biomarkers), and deconvolution biomarkers that used deconvolution methods to estimate the abundance of different cell populations in

each sample (e.g., CD8 T cells) and combined these estimates into a score (4 biomarkers). We implemented all these biomarkers using Python.

For each of the four prediction tasks (i.e., OS, PFS, 1-year death, and 6-month progression), we evaluated the performance of all transcriptomic signatures using the C-index for survival tasks and the AUC for classification tasks. We applied the same 100 cross-validation schemes as in previous experiments, focusing on the subset of 80 patients with a complete multimodal profile. For signatures that included pre-processing steps, such as standardization or PCA, these were first trained on the training set of each fold and then applied to the corresponding test set. We then compared their performance to that of the best multimodal model and the best transcriptomic model previously obtained for each task.

### Reporting summary
Further information on research design is available in the Nature Portfolio Reporting Summary linked to this article.

### Data availability
The raw data generated in this study, including PET/CT scans, digitized pathological slides, and RNA-seq profiles, are not publicly available due to patient privacy requirements. Curated clinical data are available under restricted access due to patient privacy requirements, access can be obtained upon request to Emmanuel Barillot and Nicolas Girard. Derived transcriptomic, radiomic, and pathomic features, as well as clinical outcomes (OS, PFS, and best observed RECIST response), are available at https://doi.org/10.5281/zenodo.14293431. The results from the experiments performed in this study are provided as a Source Data file. Source data are provided in this paper.

### Code availability
We have made all our codes available in GitHub repositories with associated documentation allowing for the reproduction of our multimodal analyses with external data, additional modalities, or different features (https://github.com/sysbio-curie/multipit, https://github.com/sysbio-curie/deep-multipit). The Python code to compute the 36 transcriptomic signatures and reproduce the benchmarks presented in Fig. 7 is also available on GitHub (https://github.com/sysbio-curie/tipit_benchmark_RNA). The Python code to reproduce the figures is provided in the Source Data file.

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

## Acknowledgements

We would like to thank the following collaborators at Institut Curie for their valuable support in data management and processing: A. Nicolas, R. Goudefroye, C. Martinat, M. Bouvet, and A. Vincent-Salmon from the experimental pathology platform, L. Chanas, and M. Milder from Institut Curie's Data Office, T. Ramtohul, and H. Brisse from the Department of Radiology, S. Baulande from the Next-Generation Sequencing platform, I. Bièche, and C. Callens from the Diagnostic and Theranostic Medicine Division, C. Reyes, A. Rapinat, and D. Gentien from the Genomics platform, Eugénie Genestant from the Computational Systems Biology of Cancer team, and C. Kamoun, N. Servant, and P. Hupé from the bioinformatics core facility. We also thank M. Lefevre and S. Lefranc from Institut Mutualiste Montsouris. This work was part of the TIPIT project (Towards an Integrative Approach for Precision ImmunoTherapy) funded by Fondation ARC call «SIGN'IT 2020—Signatures in Immunotherapy». The present study was also supported by the French government under the management of Agence Nationale de la Recherche as part of the 'Investissements d'avenir' program, reference ANR-19-P3IA-0001 (PRAIRIE 3IA Institute).

## Author contributions

N.C. processed the collected data, performed the analyses, implemented the computational tools, and wrote the manuscript. M.Le. designed and developed the feature extraction pipeline for pathological data and provided input on machine learning analysis. F.O. supervised the collection of PET scans and clinical data and provided input on radiomic and machine learning analysis. N.H.-B. processed segmented PET scans and provided input on radiomic analysis. M.Lu. and E.W. segmented and annotated PET scans and provided input on radiomic analysis. S.L. managed data collection and patient recruitment. P.S.F. collected and curated clinical data. C.L. estimated the TMB from the RNA-seq data and provided input on omics analysis. C.B. collected pathological slides and provided input on pathological analysis. A.Z. provided input on omics and machine learning analysis. H.S. analyzed pathological slides and provided input on the biological interpretation of predictive models. T.W. supervised the collection of pathological slides, pathological analysis, and machine learning analysis. I.B. supervised the collection of PET scans, radiomic analysis, and machine learning analysis. N.G. supervised data collection, patient recruitment, and data analysis. E.B. supervised the collection of omics data, omics analysis, and machine learning analysis. F.O., A.Z., T.W., I.B., N.G., and E.B. designed the study. N.G. and E.B. led the project. M.Le., F.O., C.L., H.S., T.W., I.B., N.G., and E.B. revised and edited the manuscript. All authors approved the manuscript.

## Competing interests

Nicolas Girard has a consulting or advisory role for the following companies: Abbvie, AMGEN, AstraZeneca, BeiGene, Bristol-Myers Squibb, Daiichi Sankyo/Astra Zeneca, Gilead Sciences, Ipsen, Janssen, LEO Pharma, Lilly, MSD, Novartis, Pfizer, Roche, Sanofi, Takeda. The other authors declare no competing interests.

## Additional information

¹Laboratoire d'Imagerie Translationnelle en Oncologie, Institut Curie, Inserm U1288, PSL Research University, Orsay, France. ²Bioinformatics and computational systems biology of cancer, Institut Curie, Inserm U900, PSL Research University, Paris, France. ³CBIO-center for Computational Biology, MINES ParisTech, PSL Research University, Paris, France. ⁴Department of medical imaging, Institut Curie, Paris, France. ⁵Department of Nuclear Medicine/PET-scan, Institut Jules Bordet, Université Libre de Bruxelles, Brussels, Belgium. ⁶Institut du Thorax Curie-Montsouris, Institut Curie, Paris, France. ⁷Department of pathology, Institut Curie, Paris, France. ⁸In silico R&D, Evotec, Toulouse, France. ⁹Immunity and cancer, Institut Curie, Inserm U932, PSL Research University, Paris, France. ¹⁰These authors contributed equally: Thomas Walter, Irène Buvat, Nicolas Girard, Emmanuel Barillot. ✉e-mail: nicolas.captier@polytechnique.org; emmanuel.barillot@curie.fr

