## [Transparent Peer Review file · Nature Communications]

Integration of clinical, pathological, radiological, and transcriptomic data improves prediction for first-line immunotherapy outcome in metastatic non-small cell lung cancer

Corresponding Author: Dr Nicolas Captier

Version 0:

Reviewer comments:

Reviewer #1

(Remarks to the Author)

The study conducted by the authors reported advancements in predictive performance achieved through the integration of multimodal data for forecasting the clinical efficacy of first-line immunotherapy. This improvement is posited in contrast to the utilization of unimodal data. However, a more thorough investigation is warranted, as several significant criticisms are outlined below.

1. The significant results must support the enhanced predictive power of multimodal data. For example, Figure 3 reveals that predictions based on multimodal data do not exhibit a statistically significant increase compared to those derived from unimodal data. Notably, predictions using RNA expression data alone demonstrate a comparable performance to the multimodal-based predictions.
2. Figure 5 raises concerns about the appropriateness of the applied T-test methodology. Within each group, data points are derived from a multitude of variables. For instance, a comparison should be undertaken based on late_LR, according to the number of applied modalities.
3. It also needs to assess whether the observed improvement attributed to multimodal data surpasses the previously established models using solely transcriptomic data. To elucidate this, a comparative analysis with existing models, such as those by Lee et al. (Cell 2021, PMID: 33857424), Jiang et al. (Nat Med. 2018, PMID: 30127393), Auslander et al. (Nat Med. 2018, PMID: 30127394), Kong et al. (Nat Comm, 2022, PMID: 35764641), and Lee et al. (Sci. Adv., 2024, PMID: 38295179) is essential. This comparative evaluation is essential to establish the effectiveness of the multimodal approach in contrast to earlier models based solely on transcriptomic data.
4. The survival and prognosis prediction framework utilizing multimodal data from cancer patients seems non-novel. Numerous prior studies have explored multimodal data integration and presented its benefits for predicting cancer patients' survival and prognosis or drug response. It raises significant concerns over the author's central argument regarding the novel use of multimodal data utilization in the introduction. Previous relevant studies, including Rami et al. (nature cancer, 2022), Lipkova et al. (cancer cell, 2022), and Wang et al. (frontiers in Bioinformatics, 2023), should be mentioned. Furthermore, they only applied existing algorithms to demonstrate the advantages of utilizing multimodal data, raising concerns about the study's novelty.
5. The results shown by the authors are nothing more than a case study. There is an urgent need to show how multimodal data can be used as a predictive biomarker.
6. The interpretability of features mentioned in line 76 needs to have adequate supporting results. Relying solely on simple feature importance analysis in the Results section needs to be more comprehensive. Further comprehensive analysis or supporting data is required. Moreover, as mentioned in lines 142-167, a rationale for features with high-importance scores is necessary. Through case studies or reports, supporting evidence associated with a correlation between such features (e.g.,

low serum albumin levels, abundant circulating neutrophils) and cancer patient survival and prognosis is essential for result reliability.

7. Clarification regarding "PD-L1 expression" as tumor proportion score (TPS) in lines 88 and 102-103 is warranted. Additionally, an explanation of the relationship between TPS of PD-L1 expression and Pembrolizumab response should be provided in advance, supported by relevant references.

8. Correlation analysis of various multimodal features is necessary. Sufficient analysis of the contribution of each multimodal data to the prediction performance of the integration model is necessary. Rather than simply stating that the multimodal approach contributes to performance, it is also essential to understand the relationship between each clinical feature.

(Remarks on code availability)

Reviewer #2

(Remarks to the Author)

In this manuscript, Captier and colleagues used multiple machine learning algorithms to integrate multimodal data derived from 317 patients with non-small cell lung cancer who received first-line immune checkpoint inhibitors, finding that multimodal strategies demonstrated improved performance for risk stratification compared to unimodal models and typical clinical features, including the current gold-standard, PD-L1 expression.

I applaud the authors for their efforts to integrate so many different data types across a large cohort. Nonetheless, the manuscript has several notable limitations:

1. One major limitation is the missingness of the data as only 80 patients have all four modalities, and those patients received a mix of immunotherapy alone and chemotherapy-immunotherapy.
2. The authors don't differentiate prognostic versus predictive biomarkers. This is especially relevant for the clinical and radiomic features, as many of the top features (e.g., albumin, ECOG performance status, metrics of tumor burden, etc) are simply prognostic.
3. While the authors observe no significant relationship between patient outcome and standard clinical biomarkers (e.g., TMB, TIL, and PD-L1 as a continuous score), it should be noted that these analyses are likely significantly underpowered.
4. Figure 1: It would be helpful to provide the numbers of risk below each Kaplan-Meier curve.
5. What assay was used for PD-L1 expression? 22C3?
6. How did the authors evaluate TIL? Was a cutoff established? Was this based upon H&E?
7. Transcriptomic data was available for 134 patients, but the authors took a very focused approach to the RNA seq data, evaluating expression of 22 oncogenes and 10 immune/stromal cells. Was the RNA sequencing limited to these genes? This is a rather modest list to evaluate, especially as various gene signatures have been correlated with response to IO in NSCLC.
8. What is included in the best clinical model (figure 6A)? It's unclear from the text and figure.
9. The authors should refrain from introduce new data in the discussion (figure s24).
10. What alterations are included in the TMB assessment? All non-synonymous alterations?

(Remarks on code availability)

As a clinical translational reviewer, I must defer to my computational colleagues for analysis of the code.

Version 1:

Reviewer comments:

Reviewer #1

(Remarks to the Author)

The author's responses to the reviewer's comments were satisfactory. I have no further inquiry.

(Remarks on code availability)

Reviewer #2

(Remarks to the Author)

The authors have addressed all of my prior comments.

(Remarks on code availability)

Reviewer #1 (Remarks to the Author):

The study conducted by the authors reported advancements in predictive performance achieved through the integration of multimodal data for forecasting the clinical efficacy of first-line immunotherapy. This improvement is posited in contrast to the utilization of unimodal data. However, a more thorough investigation is warranted, as several significant criticisms are outlined below.

1. The significant results must support the enhanced predictive power of multimodal data. For example, Figure 3 reveals that predictions based on multimodal data do not exhibit a statistically significant increase compared to those derived from unimodal data. Notably, predictions using RNA expression data alone demonstrate a comparable performance to the multimodal-based predictions.

We agree with Reviewer 1 that the differences in performance observed between unimodal and multimodal models are not always statistically significant. This could be due to several factors, including the modest sample size for testing the models (which cannot be increased in a reasonable timeframe), the noise inherent to the data and survival target, or the low statistical power of the tests comparing the performance of the two models.

However, Figure 3 and the prediction of OS, on which Reviewer 1 focused, are not the main result of our study. It is one example among many different models, predictive tasks, and integrative approaches. Figure 5 and the conclusions derived from it represent the core findings of this study: despite the limited statistical power, we observed a consistent improvement when using multiple modalities rather than a single modality. This consistency instills confidence in the benefits of multimodal strategies for predicting immunotherapy response in NSCLC and may motivate the collection of larger cohorts to gain statistical power.

Additionally, we included a patient-level comparison between the best multimodal model from Figure 3 and the RNA model (Figure 4, Lines 197-214) to go beyond the overall performance gap and show how the integration of other modalities impacts predictions. Notably, we observed that for several patients, the RNA modality misled the model toward incorrect predictions, while the radiomic and clinical modalities corrected it. This highlights that, although the multimodal benefit was not statistically significant compared to the RNA model, it reflected the positive impact of additional modalities on predictions for several patients, not just isolated cases.

2. Figure 5 raises concerns about the appropriateness of the applied T-test methodology. Within each group, data points are derived from a multitude of variables. For instance, a comparison should be undertaken based on late_LR, according to the number of applied modalities.

We thank Reviewer 1 for raising this concern, and we agree that the Student t-test may not be appropriate in this case. Instead, we used a paired sample t-test, which matches points associated with the same model across different numbers of integrated modalities. This approach tests whether the performance of multimodal approaches improves as the number of integrated modalities increases, considering that the different performance measurements are derived from the same models but with varying numbers of modalities. Figures 6 (previously Figure 5) and s15 were updated accordingly.

We intentionally displayed the performance of each model in these figures using scattered boxplots, allowing for easy visual comparison across different numbers of modalities for each model separately.

3. It also needs to assess whether the observed improvement attributed to multimodal data surpasses the previously established models using solely transcriptomic data. To elucidate this, a comparative analysis with existing models, such as those by Lee et al. (Cell 2021, PMID: 33857424), Jiang et al. (Nat Med. 2018, PMID: 30127393), Auslander et al. (Nat Med. 2018, PMID: 30127394), Kong et al. (Nat Comm, 2022, PMID: 35764641), and Lee et al. (Sci. Adv., 2024, PMID: 38295179) is essential. This

comparative evaluation is essential to establish the effectiveness of the multimodal approach in contrast to earlier models based solely on transcriptomic data.

We thank Reviewer 1 for this suggestion. We agree that our analysis would benefit from a more comprehensive comparison of unimodal and multimodal performances, particularly for the transcriptomic modality, which demonstrated good performance in our study and for which numerous signatures have been reported to be associated with immunotherapy.

To address this concern, we investigated 36 transcriptomic signatures reported in the literature as being associated with immunotherapy response, encompassing various cancer types and immune checkpoint inhibitors. Our comparison showed that our best multimodal models outperformed all these transcriptomic signatures, except in the prediction of 6-months progression, where the multimodal model outperformed 33/36 signatures. This confirmed the superiority of our multimodal approach, not only in comparison to our previously established RNA signature but also against a broad array of other RNA models described in the literature. The results of this benchmark were presented in Figure 7, and the Results (Lines 246-256) and Methods (Lines 663-679) sections were updated accordingly.

While our study did not exhaustively explore the transcriptomic modality, our primary goal was rather to assess the benefits of combining multiple modalities, even with relatively straightforward characterizations. We aimed to highlight the potential advantages of multimodal approaches and encourage future research into refining transcriptomic embeddings, while also exploring the integration of these refined signatures with other modalities, as this combination is likely to further enhance performance.

4. The survival and prognosis prediction framework utilizing multimodal data from cancer patients seems non-novel. Numerous prior studies have explored multimodal data integration and presented its benefits for predicting cancer patients' survival and prognosis or drug response. It raises significant concerns over the author's central argument regarding the novel use of multimodal data utilization in the introduction. Previous relevant studies, including Rami et al. (nature cancer, 2022), Lipkova et al. (cancer cell, 2022), and Wang et al. (frontiers in Bioinformatics, 2023), should be mentioned. Furthermore, they only applied existing algorithms to demonstrate the advantages of utilizing multimodal data, raising concerns about the study's novelty.

We agree with Reviewer 1 that other studies have already explored the benefit of multimodality in oncology. However, to our knowledge, aside from the work by Vanguri et al. (referenced as Rami et al. by Reviewer 1) which we discussed extensively in our manuscript, no study focused on the prediction of immunotherapy response for NSCLC patients using multimodal machine learning approaches. From a biomedical point of view, it seems obvious that the benefit is likely to depend on the type of cancer and the type of treatment under consideration. There is no certainty about the benefit of multimodality to predict immunotherapy response in NSCLC, and whether it is worth the effort of the tedious collection of multimodal data, or which existing machine learning methods are the most relevant for integrating multiple modalities in this specific case.

Our study addresses this gap by providing an extensive exploration of existing approaches, including the most recent ones such as the method proposed by Vanguri et al. This exploration is the novelty of our study, including investigations on a rarely explored task using original and rare multimodal data. Our findings could guide and motivate further studies on the prediction of immunotherapy response in NSCLC.

5. The results shown by the authors are nothing more than a case study. There is an urgent need to show how multimodal data can be used as a predictive biomarker.

We share Reviewer 1's concerns about the need to translate developed multimodal biomarkers into the clinical setting. However, important questions need to be addressed first: which modalities should be

integrated into multimodal biomarkers, whether multimodal biomarkers are worth it compared to unimodal ones, how to build and validate efficient multimodal biomarkers before considering their clinical application, and how to ensure their robustness against common clinical challenges such as missing modalities or low-quality data.

In our opinion, "case studies" with large and homogeneous NSCLC cohorts are still very much needed. The questions listed above must be answered before considering the application of multimodal biomarkers in a clinical context, particularly for predicting immunotherapy response in NSCLC, where very few case studies with large multimodal cohorts exist.

6. The interpretability of features mentioned in line 76 needs to have adequate supporting results. Relying solely on simple feature importance analysis in the Results section needs to be more comprehensive. Further comprehensive analysis or supporting data is required. Moreover, as mentioned in lines 142-167, a rationale for features with high-importance scores is necessary. Through case studies or reports, supporting evidence associated with a correlation between such features (e.g., low serum albumin levels, abundant circulating neutrophils) and cancer patient survival and prognosis is essential for result reliability.

We agree with Reviewer 1 that the interpretation of the learned biomarkers should be supported by comprehensive analyses, existing literature, and additional data. Although we could not access additional data due to the challenges of collecting such multimodal patient data, we made significant efforts to address the other two points: conducting comprehensive feature importance analyses and linking our findings to the literature.

First, we went beyond a simple feature importance analysis of a single top-performing model by analyzing all the learned models, incorporating multiple algorithms and predictive tasks (weighted by their performance), and highlighting consensus important features shared across all top-performing models. This comprehensive approach, supported by univariate statistical tests (Figures 2, s7–s10), strengthened our confidence in the detected associations between the top consensus features and patient outcomes.

Additionally, in the discussion, we linked the top-performing features identified by our consensus analysis with previous literature that supports their association with patient outcomes (e.g., reference 7 for clinical features, reference 23 for TMTV). To address Reviewer 2's comment 2., we included a discussion on the predictive versus prognostic nature of these features, advocating further research to distinguish between these qualities (Lines 370-377). We also added references supporting the prognostic value of top-performing clinical features, such as low serum albumin (reference 31) and abundant circulating neutrophils (reference 32).

Finally, in the discussion, we made special efforts to support the top-performing RNA features, given that RNA was the most promising modality in our analysis. In particular, we focused on the abundance of dendritic cells within the biopsy samples (Lines 322-333), citing several references (24-29) and reporting additional investigations we conducted, such as the visual detection of Tertiary Lymphoid Structures (TLS) within the pathological slides, as these structures may be associated with abundant dendritic cells. This allowed us to formulate more relevant hypotheses regarding the association of this feature with immunotherapy response.

7. Clarification regarding "PD-L1 expression" as tumor proportion score (TPS) in lines 88 and 102-103 is warranted. Additionally, an explanation of the relationship between TPS of PD-L1 expression and Pembrolizumab response should be provided in advance, supported by relevant references.

We added clarification regarding the assessment of PD-L1 expression and the computation of the Tumor Proportion Score in the first paragraph of the "Results" section (Lines 89-91).

PD-L1 TPS was initially used in clinical trials to stratify patients based on their PD-L1 expression and evaluate the efficacy of Pembrolizumab compared to chemotherapy within these subgroups. In the introduction of our manuscript, we provided references to key clinical trials that highlighted the link between PD-L1 TPS and immunotherapy benefit (references 2, 3, 4 and 15), and led to the adoption of PD-L1 TPS as a biomarker in clinical practice (reference 1). We believe these references adequately illustrate the historical association between PD-L1 TPS and Pembrolizumab response in NSCLC and justify our focus on this biomarker in our analysis, where we explore whether multimodal approaches can offer improved performance.

8. Correlation analysis of various multimodal features is necessary. Sufficient analysis of the contribution of each multimodal data to the prediction performance of the integration model is necessary. Rather than simply stating that the multimodal approach contributes to performance, it is also essential to understand the relationship between each clinical feature.

We thank Reviewer 1 for this suggestion. We conducted additional analyses to better illustrate how the different modalities correlate with each other and how they contribute to the multimodal benefit.

In supplementary Figure s11, we analyzed the correlation between the extracted features, focusing on those identified by feature importance analysis from unimodal models for predicting patient outcomes. We found that these features exhibited mild to low inter-modal correlations, supporting our initial hypothesis that different modalities capture distinct aspects of the patient's disease and that their combination may enhance the prediction of patient outcomes. The Results section of the manuscript was updated accordingly (Lines 171-174). Notably, in Figure s11.C, we used a nearest neighbors graph to display these correlations, allowing for straightforward visual inspection of the relationships between features and modalities, since feature names refer to easily understandable and interpretable characteristics.

In Figure 4, we explored the contribution of each modality to the best-performing late fusion model (combining clinical, radiomic, and RNA modalities). We particularly focused on the contributions of the clinical and radiomic modalities to better understand their impact compared to the already strong RNA model. We computed the marginal contribution at the patient level to illustrate how multimodality affects each patient and to reveal the factors behind the observed performance gains between unimodal and multimodal models. We observed that several patients were misclassified due to the RNA modality but integrating clinical and radiomic modalities corrected these predictions. We showed that features from different modalities provided contradictory information that balanced each other in the multimodal model, guiding it towards correct predictions. This analysis demonstrated that incorporating clinical and radiomic modalities changed predictions for several patients—rather than just a few isolated cases—and ultimately improved performance compared to the RNA modality alone. The Results section of the manuscript was updated accordingly (Lines 197-214).

Reviewer #2 (Remarks to the Author):

In this manuscript, Captier and colleagues used multiple machine learning algorithms to integrate multimodal data derived from 317 patients with non-small cell lung cancer who received first-line immune checkpoint inhibitors, finding that multimodal strategies demonstrated improved performance for risk stratification compared to unimodal models and typical clinical features, including the current gold-standard, PD-L1 expression.

I applaud the authors for their efforts to integrate so many different data types across a large cohort. Nonetheless, the manuscript has several notable limitations:

1. One major limitation is the missingness of the data as only 80 patients have all four modalities, and those patients received a mix of immunotherapy alone and chemotherapy-immunotherapy.

We agree with Reviewer 2 that missing modalities are one of the main limitations of our study and affect the statistical power of our analysis. However, this issue is more a limitation inherent to the task—predicting immunotherapy response for advanced NSCLC patients—than a limitation specific to our case. For instance, missing RNA data were primarily due to the low quality of the remaining biopsy material, a problem that is not easily solvable and is very likely to occur in other studies, regardless of the NSCLC cohort. Thus, our cohort, despite the limited sample size, is representative of a real-world situation. The different strategies we explored to deal with missing modalities, both at training and testing steps, are valuable for further studies with new multimodal cohorts, which will likely face the same problem of missing modalities.

We also addressed the heterogeneity associated with the mix of patients treated with immunotherapy alone and those treated with combined immunotherapy and chemotherapy by introducing an additional binary feature (i.e., 0 for immunotherapy alone and 1 for combined chemotherapy and immunotherapy) within each model that includes clinical features. This feature may help the models learn different relationships between baseline data and immunotherapy outcomes depending on the treatment received (i.e., immunotherapy alone versus combined treatment).

2. The authors don't differentiate prognostic versus predictive biomarkers. This is especially relevant for the clinical and radiomic features, as many of the top features (e.g., albumin, ECOG performance status, metrics of tumor burden, etc) are simply prognostic.

We thank Reviewer 2 for this remark. We updated the discussion to highlight that the absence of patients not treated with immunotherapy in our cohort limited our ability to distinguish between prognostic and predictive biomarkers (Lines 370-377). We still stressed that our study highlighted the potential of multimodal machine learning to develop powerful predictors of patient outcomes, also revealing interesting and promising features, such as the abundance of dendritic cells. Further studies are needed to accurately assess the predictive versus prognostic values of such predictors.

3. While the authors observe no significant relationship between patient outcome and standard clinical biomarkers (e.g., TMB, TIL, and PD-L1 as a continuous score), it should be noted that these analyses are likely significantly underpowered.

We agree that our analyses may be limited by the modest sample size of our cohort. We updated the discussion to highlight this limitation, particularly for standard clinical biomarkers. We also stressed that our comparison of different multimodal, unimodal, and standard biomarkers, using the same cohort (i.e., fixed sample size) and consistent methodologies, still provides valuable insights into the potential benefits of multimodal approaches (Lines 358-369).

4. Figure 1: It would be helpful to provide the numbers of risk below each Kaplan-Meier curve.

We thank Reviewer 2 for this suggestion. We updated Figure 1 and added the number of patients at risk and the number of observed events below each Kaplan-Meier curve. Additionally, we updated supplementary Figures s1, s2, and s3 to include these numbers.

5. What assay was used for PD-L1 expression? 22C3?

PD-L1 expression was assessed using immunohistochemistry with the SP263 and QR1 assays. This is detailed in the first paragraph of the "Results" section (Lines 89-91). Additionally, we provided further details about the Tumor Proportion Score in response to comment 7 from Reviewer 1.

6. How did the authors evaluate TIL? Was a cutoff established? Was this based upon H&E?

TILs were semi-quantitatively assessed by pathologists on routine Hematoxylin, Eosin & Saffron (HES) sections, without any cutoff, as part of routine care. This is detailed in the second paragraph of the “Results” section (Lines 112-114).

7. Transcriptomic data was available for 134 patients, but the authors took a very focused approach to the RNA seq data, evaluating expression of 22 oncogenes and 10 immune/stromal cells. Was the RNA sequencing limited to these genes? This is a rather modest list to evaluate, especially as various gene signatures have been correlated with response to IO in NSCLC.

We agree with Reviewer 2 that the extracted RNA features provide only a limited view of the transcriptomic modality, especially considering that the whole transcriptome was sequenced for the 134 patients.

We chose to use a small and simple set of interpretable transcriptomic features due to the modest sample size available for our study. Our focus was on exploring the benefits of multimodal approaches using this straightforward transcriptomic characterization, rather than conducting an exhaustive exploration of various RNA signatures and embeddings. We were concerned that testing a wide range of different signatures could lead to overfitting and overly optimistic results that might not generalize well to other datasets. Our simple selection was not arbitrary but was guided by two main considerations: the strong association between immunotherapy response and immune system activation, and the fact that RNA expression of oncogenes may provide richer information than mutational status for predicting immunotherapy response (reference 22).

We still addressed this concern, as well as Reviewer 1's concern 3., by conducting a more extensive exploration of transcriptomic signatures associated with immunotherapy response in the literature. Our results demonstrated that our multimodal models outperformed these signatures in most tasks, highlighting the superiority of our multimodal approach over the transcriptomic modality. The results of this benchmark are presented in Figure 7, and Results (Lines 246-256) and Methods (Lines 663-679) sections were updated accordingly.

8. What is included in the best clinical model (figure 6A)? It's unclear from the text and figure.

We thank Reviewer 2 for pointing out this lack of clarity. For each task, the best clinical model is the one with the lowest log-rank p-value, indicating the best stratification. For example, in the left plot of Figure 8.A (previously Figure 6.1), the best clinical model associated with the 1-year death prediction task is a perceptron that used the same 30 input clinical features as all other experiments (green bar). Therefore, the results of the feature importance analysis in Figure 2 also apply to this model.

To clarify this point, we updated the Results section of the manuscript (Lines 271-272) as well as the legend of Figure 8 (previously Figure 6). We replaced the term "best clinical model," which was misleading, with "clinical model with the lowest log-rank p-value." Additionally, we provided a more detailed explanation on how to read the left plot of Figure 8.A (previously Figure 6.A).

9. The authors should refrain from introduce new data in the discussion (figure s24).

We agree with Reviewer 2 that introducing supplementary analysis of TCGA data in the discussion could cause confusion. Therefore, we removed this analysis from the manuscript. We believe it did not add significant value compared to the potential confusion it created, especially since it focused on a cohort of NSCLC patients who were not necessarily treated with immunotherapy.

10. What alterations are included in the TMB assessment? All non-synonymous alterations?

TMB was estimated using a custom NGS panel of 571 genes, used at our institute. Only non-synonymous alterations (excluding splice site) were considered in the TMB computation. This is detailed in the "Genomic data" subsection of the "Methods" section (Lines 473-478).

We would like to highlight that two different TMB estimates were used in our analyses:

- TMB estimated from genomic data: This estimate, computed with the NGS panel and methodology described above, was used in the Kaplan-Meier analysis to evaluate the efficacy of TMB as a standard clinical biomarker (Figure 1, for 43 patients).
- TMB estimated from RNAseq data: Derived using a method from Jessen et al. 2021 and referred to as "TMB_RNA" in our analyses, this estimate was available for 110 patients. It was used as an input feature in the different predictive models that included the RNA modality.